# TULiP: Test-time Uncertainty Estimation via Linearization and Weight Perturbation

## Abstract

A reliable uncertainty estimation method is the foundation of many modern out-of-distribution (OOD) detectors, which are critical for safe deployments of deep learning models in the open world. In this work, we propose TULiP, a theoretically-driven post-hoc uncertainty estimator for OOD detection. Our approach considers a hypothetical perturbation applied to the network before convergence. Based on linearized training dynamics, we bound the effect of such perturbation, resulting in an uncertainty score computable by perturbing model parameters. Ultimately, our approach computes uncertainty from a set of sampled predictions, thus not limited to classification problems. We visualize our bound on synthetic regression and classification datasets. Furthermore, we demonstrate the effectiveness of TULiP using large-scale OOD detection benchmarks for image classification. Our method exhibits state-of-the-art performance, particularly for near-distribution samples.

## 1 Introduction

An important safety component for deep neural networks (NNs) in real-world environments is the awareness of their uncertainty upon receiving unknown or corrupted inputs. Such capability enables systems to fall back to conservative decision-making or defer to human judgments when faced with unfamiliar scenarios, which is imperative in safety-critical domains, such as autonomous driving (Atakishiyev et al., 2024) and medical applications (Esteva et al., 2017). The problem is often framed as **Out-Of-Distribution (OOD)** detection, which has witnessed significant growth in recent years (Yang et al., 2024).

Theoretically, this issue directly relates to quantifying epistemic uncertainty (Hora, 1996), which measures the lack of knowledge in a fitted model due to insufficient training data. The training process is typically modelled as a Bayesian optimization process (Wang & Yeung, 2020) with approximations for practical use (Gal & Ghahramani, 2016; Daxberger et al., 2021). More generally, epistemic uncertainty could be formalized by the variance of a trained ensemble of networks $\phi(\boldsymbol{x}; \boldsymbol{\theta})$:

$$\mathrm{Var}_{\boldsymbol{\theta}_{\mathrm{Init}}} \left[ \phi(\boldsymbol{x}; \boldsymbol{\theta}_{\mathrm{Train}}) \right], \tag{1}$$

where $\boldsymbol{\theta}_{\mathrm{Train}}$ are parameters trained by some learning algorithm from random initialization $\boldsymbol{\theta}_{\mathrm{Init}}$. Intuitively, higher prediction variance corresponds to inputs $\boldsymbol{x}$ further from training set (OOD), as there lack enough training data to eliminate model disagreements via training, hence epistemic.

Many works redesign the network or training process to be uncertainty-aware (DeVries & Taylor, 2018; Huang & Li, 2021). However, these are often impractical due to heavy computational costs, especially for large datasets. Instead, *post-hoc* methods (Liang et al., 2018; Liu et al., 2020; Hendrycks et al., 2022; Djurisic et al., 2023) are generally preferred. These approaches can be easily integrated into pre-trained models without interfering with the trained backbones, significantly enhancing their versatility (Yang et al., 2022). Nevertheless, they often lack a direct theoretical link to the training process, which weakens their theoretical foundation and necessitates extensive empirical validation.

Therefore, it is desirable to develop a post-hoc OOD method with direct theoretical justifications regarding the training process. Recent analysis of NN optimizations reveals that gradient descent can be seen as its first-order approximations (Jacot et al., 2018; Lee et al., 2019), termed *lazy* training, under specific conditions (Geiger et al., 2020). This enabled direct (but costly) computation of equation 1, as well as rigorous analysis (Kobayashi et al., 2022) and methods (He et al., 2020) on model uncertainty, even beyond the lazy regime (Chen et al., 2020).

Inspired by this series of work, we present **TULiP** (**T**est-time **U**ncertainty by **Li**nearized fluctuations via weight **P**erturbation), a post-hoc uncertainty estimator for OOD detection. Our method considers hypothetical fluctuations of the lazy training dynamics, which can be bounded under certain assumptions and efficiently estimated via weight perturbation. In practice, we found our method works well even beyond the ideal regime. Our contribution is threefold:

(i) We provide a simple, versatile theoretical framework for analyzing epistemic uncertainty at inference time in the lazy regime, which is empirically verified;

(ii) Based on our theory, we propose TULiP, an efficient and effective post-hoc OOD detector that does not require access to original training data;

(iii) We test TULiP extensively using OpenOOD (Zhang et al., 2023), a large, transparent, and unified OOD benchmark for image classifications. We show that TULiP consistently improves previous state-of-the-art methods across various settings.

The outline is as follows. Sec. 2 provides a summary of related works, Sec. 3 presents theoretical derivations, and Sec. 4 bridges theory to the implementation of TULiP. Sec. 5 reports the effectiveness of TULiP via empirical studies.

## 2 RELATED WORKS

**Uncertainty Quantification (UQ)**   As being discussed in Sec. 1, theoretically-driven methods often estimates epistemic uncertainty from a Bayesian perspective. This includes, notably, Variational Inference (Blundell et al., 2015a). Gal & Ghahramani (2016) connects Bayesian inference and the usage of Dropout layers, led their method, Monte Carlo (MC) Dropout, widely adopted in practice due to its simplicity and effectiveness. Moreover, Daxberger et al. (2021) approximates the posterior via Taylor approximation and Lakshminarayanan et al. (2017) directly used independently trained deep models as an ensemble.

**Post-hoc OOD Detectors**   For post-hoc methods, the baseline method using maximum softmax probability (MSP) was first introduced by Hendrycks & Gimpel (2017). ODIN (Liang et al., 2018) applies input preprocessing on top of temperature scaling (Guo et al., 2017a) to enhance MSP. Liu et al. (2020) proposes a simple score based on energy function (EBO). Hendrycks et al. (2022) uses maximum logits (MLS) for efficient detection on large datasets. GEN (Liu et al., 2023) adopts the generalization of Shannon Entropy, while ASH (Djurisic et al., 2023) prunes away samples' activation at later layers and simplifies the rest. Some methods also access the training set for additional information, as MDS (Lee et al., 2018b) used Mahalanobis distance with class-conditional Gaussian distributions, and ViM (Wang et al., 2022b) computes the norm of the feature residual on the principal subspace for OOD detection.

Due to the nature of post-hoc setting, most methods such as EBO, ODIN and MLS compute OOD score solely from trained models, overlooking the training process. In contrast, as previously stated, inspired by the more theoretically-aligned UQ methods, TULiP addresses the problem with regard to the training process from a theoretical aspect. In practice, TULiP works by a series of carefully constructed weight perturbations, ultimately yielding a set of model predictions, which can be seen as surrogates to posterior samples for OOD detections. Our contribution is orthogonal to methods working with logits and predictive probabilities, such as GEN, as they can work on top of TULiP outputs. In such an aspect, TULiP shares the similar plug-and-play versatility as seen in recent works, such as ReAct (Sun et al., 2021) and RankFeat (Song et al., 2022).

## 3 THEORETICAL FRAMEWORK

### 3.1 PRELIMINARIES: LINEARIZED TRAINING DYNAMICS

Jacot et al. (2018) introduced the Neural Tangent Kernel with linearization of neural networks. More importantly, they have shown that under an infinite width (lazy) limit, network parameters and hence the gradients barely change across the whole training process, justifying the linearization of the training process. Lee et al. (2019) extends the result by examining them in the parameter space, with a formal result equalizing linearized networks and empirical ones under mild assumptions.

Let $f_{\text{True}}(\boldsymbol{x}; \boldsymbol{\theta}) : \mathbb{R}^d \rightarrow \mathbb{R}^o$ be a neural network parameterized by parameters $\boldsymbol{\theta}$. The Jacobian (gradient) evaluated at $\boldsymbol{x}$ is written as $\nabla_{\boldsymbol{\theta}} f_{\text{True}}(\boldsymbol{x}) \in \mathbb{R}^{o \times |\boldsymbol{\theta}|}$, where $|\boldsymbol{\theta}|$ is the cardinality of $\boldsymbol{\theta}$, i.e., the number of parameters in the network.

Let $f(\boldsymbol{x}; \boldsymbol{\theta})$ denote the network linearized at $\boldsymbol{\theta}^*$:

$$f(\boldsymbol{x}; \boldsymbol{\theta}) := f_{\text{Init}}(\boldsymbol{x}) + \nabla_{\boldsymbol{\theta}} f_{\text{True}}(\boldsymbol{x})|_{\boldsymbol{\theta}=\boldsymbol{\theta}^*} (\boldsymbol{\theta} - \boldsymbol{\theta}^*), \tag{2}$$

where $f_{\text{Init}}(\boldsymbol{x})$ is the initial network function. Typically, the network is linearized at initialization $\boldsymbol{\theta}^* = \boldsymbol{\theta}_{\text{Init}}$. Here, we treat it as a linear approximation to the true training dynamics. For our convenience, we will interchangeably use $\nabla_{\boldsymbol{\theta}} f(\boldsymbol{x})$ as $\nabla_{\boldsymbol{\theta}} f_{\text{True}}(\boldsymbol{x})|_{\boldsymbol{\theta}=\boldsymbol{\theta}^*}$.

We consider the training data $\boldsymbol{x}$ within an empirical dataset $X$. For a twice-differentiable loss function $\ell(f(\boldsymbol{x}); y(\boldsymbol{x}))$ with target $y(\boldsymbol{x})$, we write it's gradient w.r.t. $f(\boldsymbol{x})$ as $\ell'(f(\boldsymbol{x}); y(\boldsymbol{x}))$ (or simply $\ell'(f(\boldsymbol{x}))$). Then, following Lee et al. (2019), $f$ is trained on $X$ following the gradient flow:

$$\partial_t f_t(\boldsymbol{x}) = -\eta \mathbb{E}_{\boldsymbol{x}'} \left[ \Theta(\boldsymbol{x}, \boldsymbol{x}') \ell'(f_t(\boldsymbol{x}'); y(\boldsymbol{x}')) \right], \tag{3}$$

where $\mathbb{E}_{\boldsymbol{x}'}$ is the expectation w.r.t. the empirical distribution for $\boldsymbol{x}' \in X$, $\eta$ is the learning rate and $f_t$ denotes the network $f$ at time $t \in [0, T]$. Given inputs $\boldsymbol{x}, \boldsymbol{x}'$, The Neural Tangent Kernel (NTK) $\Theta(\boldsymbol{x}, \boldsymbol{x}') \in \mathbb{R}^{o \times o}$ defined as $\Theta(\boldsymbol{x}, \boldsymbol{x}') := \nabla_{\boldsymbol{\theta}} f(\boldsymbol{x}) \nabla_{\boldsymbol{\theta}} f(\boldsymbol{x}')^\top$ governs the linearized training equation 3. Under the lazy limit, the NTK $\Theta(\boldsymbol{x}, \boldsymbol{x}')$ stays constant across the training process and hence is independent of $t$. Hereon, we assume the unique existence of the solution to equation 3.

**Notations** Let $\boldsymbol{z} \in \mathbb{R}^d$ be an arbitrary test point. Let $\| \cdot \|$ denote the Euclidean norm and induced 2-norm for vectors and matrices. Let $\| \cdot \|_{\text{F}}$ denote the matrix Frobenius norm. We also denote $\|\cdot\|_X := \mathbb{E}_{\boldsymbol{x}} \left[ \| \cdot \|^2 \right]^{1/2}$ the data-dependent norm through out the following descriptions. Finally, let $f(\boldsymbol{z}) \lesssim g(\boldsymbol{z})$ indicate $f(\boldsymbol{z}) \leq K g(\boldsymbol{z}) + M$, up to some constant $K, M$ independent of $\boldsymbol{z}$.

## 3.2 MODELING UNCERTAINTY

Under our problem setting, neither the distribution of initialized models nor the training process is accessible, which renders a significant difficulty for the direct computation of the uncertainty shown in equation 1. Instead, we choose to intuitively model it by considering a perturbation applied towards the network function $f(\boldsymbol{x})$, at a time $t = t_s$ *before* the training terminates at $t = T$. This perturbation prior to convergence is *hypothetical*, as it is inaccessible in our post-hoc setting, and we will only use it to establish our theoretical framework.

Formally, consider a perturbation to $f_{t_s}$ at $t = t_s$ as $\hat{f}_{t_s}(\boldsymbol{x}) = f_{t_s}(\boldsymbol{x}) + \Delta f(\boldsymbol{x})$. After the perturbation, the perturbed network $\hat{f}(\boldsymbol{x})$ will be trained following the same dynamics as equation 3:

$$\partial_t \hat{f}_t(\boldsymbol{x}) = -\eta \mathbb{E}_{\boldsymbol{x}'} [\Theta(\boldsymbol{x}, \boldsymbol{x}') \ell'(\hat{f}_t(\boldsymbol{x}'); y(\boldsymbol{x}'))], \tag{4}$$

until termination time $T$.

Under such a perturb-then-train process, we model the epistemic uncertainty as the difference between converged networks, reads $\|f_T(\boldsymbol{z}) - \hat{f}_T(\boldsymbol{z})\|$. It measures the fluctuation of the training process, capturing the sensitivity of training w.r.t. noise. Indeed, by applying a perturbation at $t = 0$, we essentially perturb $f_{\text{Init}}$, which can be seen as a sampling process from some model prior (Appendix A.7). Therefore, $\hat{f}_T(\boldsymbol{z})$ can be interpreted as samples from the trained ensemble as in equation 1, where their variance reflects epistemic uncertainty.

However, as stated above, in practice we only know the trained network $f_T$ at $t = T$. It would be impractical to recover the full training trajectory, apply the perturbation at $t = t_s$ and then retrain the network. Therefore, in the following, we will come up with a bound of $\|f_T(\boldsymbol{z}) - \hat{f}_T(\boldsymbol{z})\|$ given the strength of the perturbation $\Delta f$, which can be evaluated at $\boldsymbol{z}$ without actually retrain the network. Thus, the perturbation is *hypothetical*, as it has never been applied in our practice.

We first present this bound, then we examine a method to estimate the bound without explicit access to training data.

## 3.3 BOUNDING LINEARIZED TRAINING FLUCTUATIONS

We shall introduce the following assumptions:

A1. (Boundedness) For $t \in [0, T]$, $f(\boldsymbol{x})$, $\nabla_{\boldsymbol{\theta}} f(\boldsymbol{x})$, $\ell$ and $\ell'$ stay bounded, uniformly on $\boldsymbol{x}$.

A2. (Smoothness) Gradient $\ell'$ of loss function $\ell$ is Lipschitz continuous: $\forall x \in X$; $\|\ell'(\hat{y}; y(\boldsymbol{x})) - \ell'(\hat{y}'; y(\boldsymbol{x}))\| \leq L\|\hat{y} - \hat{y}'\|$.

A3. (Perturbation) The perturbation $\Delta f$ can be uniformly bounded by a constant $\alpha$, that is, for all $\boldsymbol{x}$ (not limited to the support of training data), i.e., $\forall x \in \mathbb{R}^d$; $\|\Delta f(\boldsymbol{x})\| \leq \alpha$.

A4. (Convergence) Finally, for the original network trained via equation 3 and the perturbed network trained via equation 4, we assume *near-perfect convergence* on the training set $\boldsymbol{x}$ at termination time $t = T$, i.e., $\exists \beta \in \mathbb{R}, \forall x \in X$; $\|f_T(\boldsymbol{x}) - \hat{f}_T(\boldsymbol{x})\| \leq \beta$.

Under reasonable conditions, it has been shown both empirically (Zhang et al., 2017) and theoretically (Du et al., 2019) that overparameterized NNs trained via SGD is able to achieve near-zero training loss on almost arbitrary training sets. To nice loss functions as $\ell(\boldsymbol{y}; \boldsymbol{y}') = 0$ implies $\boldsymbol{y} = \boldsymbol{y}'$, this implies A4.

Jacot et al. (2018) connected lazy NNs trained with mean square error (MSE) loss and kernel ridge regression. Essentially, it hints that under such a setup, an NN embeds datapoints $\boldsymbol{x}$ into gradients $\nabla_{\boldsymbol{\theta}} f(\boldsymbol{x})$. Indeed, for a general class of loss functions, it is possible to show that:

**Theorem 3.1.** *Under assumptions A1-A4, for a network $f$ trained with equation 3 and a perturbed network $\hat{f}$ trained with equation 4, the perturbation applied at time $t_s = T - \Delta T$ bounded by $\alpha$, we have*

$$\|f_T(\boldsymbol{z}) - \hat{f}_T(\boldsymbol{z})\| \leq \inf_{\boldsymbol{x} \in X} C\|\nabla_{\boldsymbol{\theta}} f(\boldsymbol{z}) - \nabla_{\boldsymbol{\theta}} f(\boldsymbol{x})\|_{\mathrm{F}} + 2\alpha + \beta, \tag{5}$$

*where $C = \frac{\alpha\eta\bar{\Theta}_X^{1/2}}{\lambda_{max}} \left(e^{(T-t_s)L\lambda_{max}} - 1\right)$, $\bar{\Theta}_X^{1/2} := \|\nabla_{\boldsymbol{\theta}} f(\boldsymbol{x})\|_X$ is average gradient norm over training data, and $\lambda_{max} := \frac{1}{\sqrt{N}}\|\boldsymbol{G}\|$ for a generalized Gram matrix $G_{i,j} := \|\Theta(x_i, x_j)\|$ of dataset $X = \{x_1, x_2, \ldots, x_N\}$.*

*Proof.* With an arbitrarily chosen pivot point $\boldsymbol{x}^*$ from the training set, it is possible to bound $\|f(\boldsymbol{z}) - f(\boldsymbol{x}^*)\|$ and $\|\hat{f}(\boldsymbol{z}) - \hat{f}(\boldsymbol{x}^*)\|$ by bounding the fluctuations on the training set. The theorem then follows from assumption A4. Please check Sec. A.3 for details. $\square$

We see that the bound on the training fluctuation is dominated by the distance from test point $\boldsymbol{z}$ to the training set $X$ in the "embedding space" of gradients. Expanding it with $\|\boldsymbol{A}\|_{\mathrm{F}} = \left(\mathrm{Tr}(\boldsymbol{A}\boldsymbol{A}^{\top})\right)^{1/2}$ (detailed in Sec A.3), we can observe its connection with the NTK $\Theta$:

$$\inf_{\boldsymbol{x} \in X} \|\nabla_{\boldsymbol{\theta}} f(\boldsymbol{z}) - \nabla_{\boldsymbol{\theta}} f(\boldsymbol{x})\|_{\mathrm{F}} = \inf_{\boldsymbol{x} \in X} \left[\mathrm{Tr}\left(\Theta(\boldsymbol{z}, \boldsymbol{z}) + \Theta(\boldsymbol{x}, \boldsymbol{x}) - 2\Theta(\boldsymbol{z}, \boldsymbol{x})\right)\right]^{1/2}. \tag{6}$$

### 3.4 ESTIMATING THE BOUND WITHOUT TRAINING DATA

However, given no access to training data, the term $\nabla_{\boldsymbol{\theta}} f(\boldsymbol{x})$ is intractable. Moreover, even with full access, computing the minimum of equation 6 requires significant computational effort. A typical training dataset often contains millions of data points. Besides, given the size of the network, storing the full gradient for a single data point may already require significant memory.

Fortunately, we might be able to recover some information about the training dataset $\boldsymbol{x}$ from the parameters $\boldsymbol{\theta}_T$ and $\boldsymbol{\theta}_{t_s}$, given them being trained on the dataset via lazy gradient descent:

**Lemma 3.2.** *We assume the lazy regime for the training process, i.e., the NTK, $\Theta(\boldsymbol{z}, \boldsymbol{x})$, does not depend on the parameter $t \in [t_{t_s}, T]$. Under assumption A1, with the model parameters $\boldsymbol{\theta}_T$ trained from $\boldsymbol{\theta}_{t_s}$ with equation 3 over the training set $\boldsymbol{x}$ and $t_s < T$, we have:*

$$\|\nabla_{\boldsymbol{\theta}} f_T(\boldsymbol{z})(\boldsymbol{\theta}_T - \boldsymbol{\theta}_{t_s})\| \leq K \cdot \mathrm{Tr}\left(\mathbb{E}_{\boldsymbol{x}}\left[|\Theta_T(\boldsymbol{z}, \boldsymbol{x})|\right]\right), \tag{7}$$

*for some $K$ independent of $\boldsymbol{z}$. $|\Theta_T(\boldsymbol{z}, \boldsymbol{x})|$ is defined as the unique symmetric positive semi-definite solution of $|\Theta_T(\boldsymbol{z}, \boldsymbol{x})|^2 = \Theta_T(\boldsymbol{z}, \boldsymbol{x})^{\top}\Theta_T(\boldsymbol{z}, \boldsymbol{x})$. It is an extension of absolute values to matrices.*

*Proof.* We prove the lemma under the weakly lazy regime, i.e., we allow the weak dependency of $\Theta_t$ on $t$. The consequence follows from the Hölder's inequality. Please check Sec. A.4 for details. $\square$

We further introduce an assumption on the closeness between the test point $z$ and the dataset $X$, such that equation 6 can be bounded by

$$\inf_{\boldsymbol{x} \in X} \text{Tr}\left(\Theta(\boldsymbol{z}, \boldsymbol{z}) + \Theta(\boldsymbol{x}, \boldsymbol{x}) - 2\Theta(\boldsymbol{z}, \boldsymbol{x})\right) \leq \text{Tr}\left(\Theta(\boldsymbol{z}, \boldsymbol{z}) + \mathbb{E}_{\boldsymbol{x}}\left[\Theta(\boldsymbol{x}, \boldsymbol{x})\right] - 2\mathbb{E}_{\boldsymbol{x}}\left[|\Theta(\boldsymbol{z}, \boldsymbol{x})|\right]\right). \quad (8)$$

Intuitively, if across all $\boldsymbol{x} \in X$, $\text{Tr}\left(\Theta(\boldsymbol{z}, \boldsymbol{x})\right)$ only attains a small negative value within a limited subset of $\boldsymbol{x}$, and $\sup_{\boldsymbol{x} \in X} \text{Tr}\left(\Theta(\boldsymbol{z}, \boldsymbol{x})\right)$ is largely positive ($\boldsymbol{z}$ is close to $\boldsymbol{x}$ in the sense of $\Theta$), equation 8 holds. Fig. 1 d) provides empirical justifications for this closeness assumption.

With the closeness assumption, we can then derive a bound on equation 5 using Lemma 3.2:

**Proposition 3.3.** *Given $\boldsymbol{z}$ and $\boldsymbol{x}$ satisfies equation 8, equation 5 can be further upper-bounded by:*

$$\|f_T(\boldsymbol{z}) - \hat{f}_T(\boldsymbol{z})\| \lesssim \left[\text{Tr}\left(\Theta(\boldsymbol{z}, \boldsymbol{z}) + \mathbb{E}_x[\Theta(\boldsymbol{x}, \boldsymbol{x})]\right) - 2K \left\|\nabla_{\boldsymbol{\theta}} f_T(\boldsymbol{z})\left(\boldsymbol{\theta}_T - \boldsymbol{\theta}_{t_s}\right)\right\|\right]^{1/2}, \quad (9)$$

*for some $K$ independent from $\boldsymbol{z}$ (that may differ from the one in Lemma 3.2).* $\qquad\square$

Given test point $\boldsymbol{z}$ and parameters $\boldsymbol{\theta}_T, \boldsymbol{\theta}_{t_s}$, equation 9 provides a tractable bound for equation 5.

**Network Ensemble** We close this section by the fact that

$$\text{Tr}(\text{Var}_{\Delta f}[\hat{f}_T(\boldsymbol{z})]) \leq \mathbb{E}_{\Delta f}[\|\hat{f}_T(\boldsymbol{z}) - f_T(\boldsymbol{z})\|^2], \quad (10)$$

which can be then bounded by equation 9. As we stated before, $\hat{f}_T(\boldsymbol{z})$ can be seen as samples from the trained ensemble as in equation 1. In practice, it is often beneficial to obtain such samples. In the next section, we will present a heuristic method to estimate $\hat{f}_T(\boldsymbol{z})$ by matching variances.

## 4 IMPLEMENTATION

In this section, we present the key implementation strategies that enhance the practical effectiveness of our method, TULiP, summarized in Alg. 1. We elaborate on its design in the following subsections by referring to lines in Alg. 1.

In contrast to the linearized network $f(\boldsymbol{x}; \boldsymbol{\theta})$, let $f_t^{emp}(\boldsymbol{x}; \boldsymbol{\theta})$ denote a network trained empirically. Intuitively, trajectories of $f_t(\boldsymbol{x}; \boldsymbol{\theta})$ and $f_t^{emp}(\boldsymbol{x}; \boldsymbol{\theta})$ is similar when $\boldsymbol{\theta}^* = \boldsymbol{\theta}_{\text{Init}}$ with a small learning-rate (Lee et al., 2019; Geiger et al., 2020). Under a post-hoc setting, as only converged models are available, we take $t_s = 0$ and substitute $\boldsymbol{\theta}_{t_s}$ with $\mathbb{E}[\boldsymbol{\theta}_0] = \boldsymbol{0}$ (or other mean specified by initialization schemes) in our implementation.

We first introduce how we estimate equation 9 using $f_T^{emp}$ at $t = T$. Then, we introduce the construction of surrogate posterior samples that greatly enhance our method.

### 4.1 LAYER-WISE SCALING (LINE 2 - 6)

Lazy training often fails to capture the full characteristics of practically trained neural networks (Seleznova & Kutyniok, 2022). In our experiments, we have observed significant changes in the empirical NTK throughout the training process. Therefore, to better capture a full picture of the whole training trajectory with only $f_T^{emp}$, we propose to use a reweighted empirical NTK to approximate the kernel $\Theta$ used for linearization in equation 3 and beyond:

$$\nabla_{\boldsymbol{\theta}} f_T^{emp}(\boldsymbol{z}) \boldsymbol{\Gamma}^2 \nabla_{\boldsymbol{\theta}} f_T^{emp}(\boldsymbol{x})^\top \approx \Theta(\boldsymbol{z}, \boldsymbol{x}), \quad (11)$$

where $\boldsymbol{\Gamma}$ is a diagonal matrix of size $|\boldsymbol{\theta}| \times |\boldsymbol{\theta}|$ that shares the same value for parameters within the same layer. Similarly, $\nabla_{\boldsymbol{\theta}} f_T^{emp}(\boldsymbol{z}) \boldsymbol{\Gamma} \approx \nabla_{\boldsymbol{\theta}} f(\boldsymbol{z})$.

This reweighting is applied as a layer-wise scaling over the empirical NTK evaluated at convergence. Given layer $l$ with parameters $\boldsymbol{\theta}_l$, we scale them as

$$\boldsymbol{\Gamma}_l := (1/\sqrt{|\boldsymbol{\theta}_l|}) \cdot \boldsymbol{I}, \quad (12)$$

where $\boldsymbol{\Gamma}_l$ is the diagonal entries in $\boldsymbol{\Gamma}$ corresponds to $\boldsymbol{\theta}_l$. We note that such scaling is highly heuristical, and we adopted it for its simplicity (further discussed in Appendix C.2). For converged networks, such scaling could potentially recover an earlier network state, which is more representative of the training trajectory as the majority of training has been done in this stage (in the sense of raw performance, e.g., accuracy). We demonstrate this effect empirically in Fig. 1.

In practice, to apply layer-wise scaling, we can simply multiply $\boldsymbol{\Gamma}$ to the perturbations introduced below.

---

**Algorithm 1** TULiP for Classifiers. $\circ$: Elementwise product

---

**Input**: Input $z \in \mathbb{R}^d$, trained parameters $\boldsymbol{\theta}_T$, network $f^{emp}(z; \boldsymbol{\theta})$
**Parameter**: Perturbation strength $\epsilon, \delta$; Parameter $\lambda$; Number of posterior samples $M$
**Output**: Uncertainty score $U$

1: $\boldsymbol{\theta}_{t_s} \leftarrow \mathbf{0}$
2: **for all** Layer $l$ of $f^{emp}$ **do**                                 $\triangleright$ Layer-wise scaling
3:     $\boldsymbol{\theta}_l \leftarrow$ parameters of layer $l$ from $\boldsymbol{\theta}_T$
4:     $\Gamma_l \leftarrow \mathbf{1}(1/\sqrt{|\boldsymbol{\theta}_l|})$                         $\triangleright$ Vector of length $|\boldsymbol{\theta}_l|$
5: **end for**
6: $\Gamma \leftarrow \text{Concatenate}(\Gamma_l)$
7: **for** $i = 1, \ldots, M$ **do**
8:     Sample $v_i \in \mathbb{R}^{|\boldsymbol{\theta}|}$ from $\mathcal{N}(0, \epsilon^2 \boldsymbol{I})$
9:     $\tilde{f}_i^{raw}(\boldsymbol{z}) \leftarrow f^{emp}(z; \boldsymbol{\theta}_T + \Gamma \circ v_i)$
10: **end for**
11: $\widetilde{\Theta}_{\mathrm{Tr}}(\boldsymbol{z}, \boldsymbol{z}) \leftarrow \frac{1}{M} \sum_i \|\tilde{f}_i^{raw}(\boldsymbol{z}) - f^{emp}(\boldsymbol{z}; \boldsymbol{\theta}_T)\|^2$         $\triangleright$ Estimation of $\mathrm{Tr}\,\Theta(\boldsymbol{z}, \boldsymbol{z})$
12: $D \leftarrow \|f^{emp}(\boldsymbol{z}; \boldsymbol{\theta}_T + \epsilon\delta\Gamma \circ (\boldsymbol{\theta}_T - \boldsymbol{\theta}_{t_s})) - f^{emp}(\boldsymbol{z}; \boldsymbol{\theta}_T)\|$
13: $S \leftarrow \widetilde{\Theta}_{\mathrm{Tr}}(\boldsymbol{z}, \boldsymbol{z}) - \lambda D$             $\triangleright$ Estimation of equation 9 up to $\mathbb{E}_{\boldsymbol{x}}[\Theta(\boldsymbol{x}, \boldsymbol{x})]$ and square root
14: $\gamma \leftarrow \sqrt{\max(S, 0)/\widetilde{\Theta}_{\mathrm{Tr}}(\boldsymbol{z}, \boldsymbol{z})}$
15: **for** $i = 1, \ldots, M$ **do**                                 $\triangleright$ Surrogate posterior samples
16:     $\tilde{f}_i(\boldsymbol{z}) \leftarrow (1 - \gamma)f^{emp}(z; \boldsymbol{\theta}_T) + \gamma\tilde{f}_i^{raw}(\boldsymbol{z})$
17: **end for**
18: $U \leftarrow \mathbb{H}_y(\frac{1}{M}\sum_i \text{softmax}(\tilde{f}_i(\boldsymbol{z})))$

---

### 4.2 ESTIMATION OF JACOBIAN (LINE 7 - 13)

Estimating gradients explicitly is both time and memory-consuming, especially for networks with large output dimensions. Fortunately, for Jacobian-vector products as in equation 9, we may use a first-order approximation to avoid computing the gradients with a backward pass:

$$\lim_{\delta \to 0} \frac{1}{\delta} \left( f^{emp}(z; \boldsymbol{\theta}_T + \delta\boldsymbol{\Gamma}\tilde{\boldsymbol{\theta}}) - f^{emp}(z; \boldsymbol{\theta}_T) \right) \approx \nabla_{\boldsymbol{\theta}} f_T(\boldsymbol{z})\tilde{\boldsymbol{\theta}}. \tag{13}$$

We use it in line 12 of Alg. 1 to estimate $\|\nabla_{\boldsymbol{\theta}} f_T(\boldsymbol{z})(\boldsymbol{\theta}_T - \boldsymbol{\theta}_{t_s})\|$ with $D$ up to multiplications.

For $\mathrm{Tr}\,\Theta(\boldsymbol{z}, \boldsymbol{z})$, we could estimate its value with Hutchinson's Trace Estimator (Avron & Toledo, 2011) (line 7-11).

**Proposition 4.1.** *Suppose that $f^{emp}$ is $\gamma$-smooth w.r.t. $\boldsymbol{\theta}$, i.e.,*

$$\|\nabla_{\boldsymbol{\theta}} f^{emp}(\boldsymbol{z}; \boldsymbol{\theta}) - \nabla_{\boldsymbol{\theta}} f^{emp}(\boldsymbol{z}; \boldsymbol{\theta}')\|_{\mathrm{F}} \leq \gamma\|\boldsymbol{\theta} - \boldsymbol{\theta}'\|.$$

*Let $\mathbf{v}$ be a random variable such that $\mathbb{E}_{\mathbf{v}}[\mathbf{v}] = \mathbf{0}, \mathbb{E}_{\mathbf{v}}[\mathbf{v}\mathbf{v}^{\top}] = \epsilon^2 \boldsymbol{I}$ and $\mathbb{E}_{\mathbf{v}}[\|\mathbf{v}\|^k] \leq C_k\epsilon^k$ for $k = 3, 4$, where $C_k$ is a constant depending on $k$ and the dimension of $\mathbf{v}$. Then, under A1, it holds that*

$$\lim_{\epsilon \to 0} \frac{1}{\epsilon^2}\mathbb{E}_{\mathbf{v}}\left[\|f^{emp}(\boldsymbol{z}; \boldsymbol{\theta}_T + \boldsymbol{\Gamma}\mathbf{v}) - f^{emp}(\boldsymbol{z}; \boldsymbol{\theta}_T)\|^2\right] = \mathrm{Tr}\left(\nabla_{\boldsymbol{\theta}} f^{emp}(\boldsymbol{z}; \boldsymbol{\theta}_T)\boldsymbol{\Gamma}^2\nabla_{\boldsymbol{\theta}} f^{emp}(\boldsymbol{z}; \boldsymbol{\theta}_T)^{\top}\right). \tag{14}$$

Note that the multi-dimensional normal distribution with mean zero and variance-covariance matrix $\epsilon^2 \boldsymbol{I}$ agrees to the condition of $\mathbf{v}$. Proposition 4.1 and the approximation equation 11 ensures that $\mathrm{Tr}\,\Theta(\boldsymbol{z}, \boldsymbol{z})$ is approximated by $\epsilon^{-2}\mathbb{E}_{\mathbf{v}}[\|f^{emp}(\boldsymbol{z}; \boldsymbol{\theta}_T + \boldsymbol{\Gamma}\mathbf{v}) - f^{emp}(\boldsymbol{z}; \boldsymbol{\theta}_T)\|^2]$ with a small $\epsilon$.

*Proof.* Please check Sec. A.5 for details. $\square$

From above, $\boldsymbol{z}$-relevant terms in equation 9 can be approximated while avoiding explicit computation of $\nabla_{\boldsymbol{\theta}} f(\boldsymbol{z})$. Specifically, in line 13, $S$ provides an estimation of the upper-bound equation 9 up to $\mathbb{E}_{\boldsymbol{x}}[\Theta(\boldsymbol{x}, \boldsymbol{x})]$, square root and multiplicative constants. Here, the hyper-parameter $\lambda$ acts as a proxy to the constant $K$ in Lemma 3.2. Such approximation is implemented by perturbations to $\boldsymbol{\theta}$, thus compatible with mini-batching, enabling fast computation with $\mathcal{O}(M)$ forward passes.

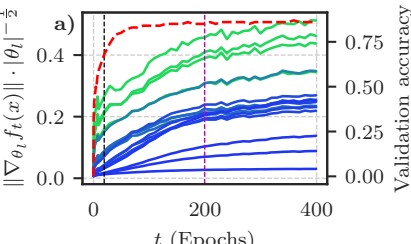 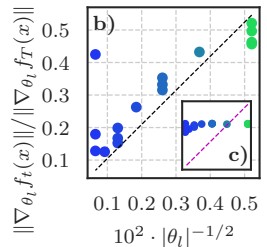 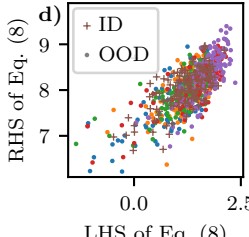

Figure 1: **a) b) c)** Empirical justifications for the layer-wise scaling scheme. We trained a ResNet-18 on CIFAR-10 dataset with SGD-momentum for 400 epochs. **a)** Average magnitude of Jacobian entries for different *conv* layers (solid) vs. time vs. validation accuracy (dashed). Layers with more parameters (lower) train slower compared to layers with fewer parameters (upper). **b)** The ratio between Jacobian norm at $t = $ Epoch 20 and $T = $ Epoch 400 vs. our scaling equation 11. A proportional relationship (dashed) supports such scaling in recovering an early NTK critical to training. **c)** Such relationship disappears at $t = $ Epoch 200. **d)** Verification for equation 8 (detailed setups in Sec. 5). ID Dataset: ImageNet-1K. OOD Datasets: ImageNet-C, ImageNet-R, SSB-Hard (Vaze et al., 2022), iNaturalist (Van Horn et al., 2018), Textures (Cimpoi et al., 2014).

### 4.3 SURROGATE NETWORK ENSEMBLE (LINE 14 - 18)

As stated in Sec. 3, the bound in equation 5 or 9 is insufficient to capture a full picture – for example, a well-trained classifier can be certain that a test data belongs to neither class, in a sense that an evaluation of equation 9 yields a small value, but their prediction (i.e., it belongs to neither classes) $\hat{f}_T(z)$ indicates OOD input. Informally yet intuitively, $\hat{f}_T(z)$ can also be seen as predictions of models sampled from some model posterior.

To this end, we propose to approximate $\hat{f}_T(z)$ by constructing $\tilde{f}(z)$, via the process described in line 14-18 of Alg. 1. In short, we squeeze the perturbed predictions $\tilde{f}_i^{raw}(z)$, producing $\tilde{f}(z)$, so that their variance matches equation 10, which is an upper bound of the variance of true perturbed predictions $\mathrm{Tr}(\mathrm{Var}_{\Delta_f}[\hat{f}_T(z)])$.

From line 16, it is possible to show

$$\mathrm{Tr}(\mathrm{Var}_i[\tilde{f}_i(z)]) \approx \gamma^2 \cdot \mathrm{Tr}(\mathrm{Var}_i[\tilde{f}_i^{raw}(z)]) \ = S, \tag{15}$$

for a positive $S$ and small $\epsilon$ such that $\mathbb{E}_i[\tilde{f}_i^{raw}(z)] \approx f^{emp}(z; \theta_T)$. Note that $\gamma$ is given in line 14 of Alg. 1, and $S$ is an estimation of equation 9 as stated in the previous subsection. Sec. A.6 provides additional derivations to clarify their relationships.

For classification problems, after obtaining $\tilde{f}(z)$, it is then able to combine the epistemic uncertainty and model prediction. One common approach is the Information Entropy $\mathbb{H}$ (Shannon, 1948) of the mean prediction: $\mathbb{H}_y[\tilde{f}(z)] := -\sum_{y=1}^o \mathbb{E}[\sigma(\tilde{f}(z))]_y \log \mathbb{E}[\sigma(\tilde{f}(z))]_y$, where $\sigma$ is the softmax operation producing the class probabilities and $[\cdot]_y$ takes the $y$-th component from a vector. Other methods, such as GEN (Liu et al., 2023), can also be naturally incorporated to TULiP by replacing line 18 in Alg. 1.

Yet, significant simplifications have been made for computational clarity. For example, $\mathbb{E}_x[\Theta(x, x)]$ has been omitted as it is intractable and irrelevant to $z$. Empirically we have found that our method is effective despite such simplifications, which will be demonstrated in the next section. We choose to simplify this for clarity, avoiding the introduction of new hyper-parameters to TULiP.

Alg. 1 summarizes TULiP, our proposed uncertainty estimator for OOD detection. Although Alg. 1 gives TULiP for classification, it naturally generalizes to non-classification problems as TULiP constructs surrogate posterior samples.

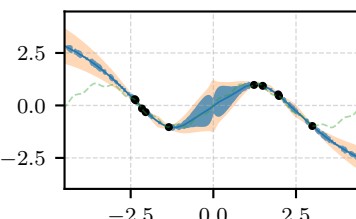 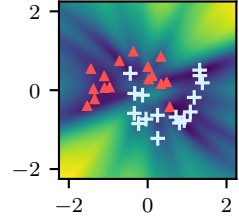 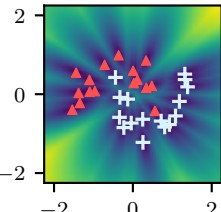

Figure 2: Verification of Thm. 3.1 with synthetic data. From left to right, **a)**: Regression on Splines. Light shade: the bound equation 5, heavy shade: Ground-truth ensemble (equation 1), black dots: training data. **b) c)**: Binary classification on Two-Moons. The brighter colour indicates larger values across the input space. **b)**: Prediction variance of 20 simulated runs, **c)**: Evaluation of equation 5.

## 5 EXPERIMENTS

### 5.1 EMPIRICAL VALIDATION FOR SECTION 3

**Synthetic Datasets** We begin this section by validating the original bound presented in equation 5. Two types of artificial datasets have been considered: namely *Splines* for regression and *Two-Moons* for classification problems. A 3-layer infinite-wide feed-forward neural network is used and we solved the lazy training dynamics over the dataset using the *neural-tangents* library (Novak et al., 2020). For Splines, we used MSE loss and computed the exact Gaussian ensemble (Lee et al., 2019). For Two-Moons, we used binary cross-entropy loss and numerically simulated the lazy gradient descent for 20 runs. Results are shown in Fig. 2. It suggests that our bound equation 5 based on training fluctuations is able to capture the true epistemic uncertainty as in equation 1, justifies further developments of our method.

**Closeness Condition** We proceed by presenting empirical justifications for equation 8 in Fig. 1 d). We used a ResNet18 (He et al., 2016) pre-trained on ImageNet-1K (Russakovsky et al., 2015), computed equation 8 by 256 samples from the ID dataset (ImageNet-1K) and 128 samples per OOD dataset. The scaled empirical NTK as in equation 11 is used in this experiment. Clearly, we see that equation 8 is satisfied by a large margin under this practical setting.

### 5.2 OUT-OF-DISTRIBUTION DETECTION

In this subsection, we demonstrate the effectiveness of our method for OOD detection in real-world scenarios by comparing TULiP with state-of-the-art OOD detectors.

**Experiment Setup** We evaluate the performance of TULiP with OOD detection tasks based on manually defined ID-OOD dataset pairs (Zhang et al., 2023). For TULiP, we use $M = 10$ surrogate posterior samples with $\epsilon = 2.0$, $\delta = 2$ and $\lambda = \sqrt{o}$ where $o$ is the number of output dimensions. Only weights in the convolutional and fully connected layers are being perturbed, while biases are ignored. Following Zhang et al. (2023), we conduct a hyper-parameter search on a small validation set whenever possible, within a reasonable range of $\epsilon \in \{0.1, 0.5, 1.5, 2.0\}$, $\delta \in \{2, 5, 8\}$ and $\lambda \in \{\sqrt{o}, 3\sqrt{o}\}$. We explain our choice for hyper-parameters in Sec. B.2. We consider two OOD scenarios, namely Semantic-Shift OOD (SS-OOD) and Covariate-Shift OOD (CS-OOD) (Yang et al., 2024). The fundamental difference between them is that SS-OOD considers distributional shift on both input $x$ and label $y$, often with unseen classes. CS-OOD considers distributional shift solely on input $x$. Recently, Yang et al. (2021) raised concerns regarding the negligible covariate shifts between ID and OOD data with same labels. Our setup does not contradict with this work as overlapping classes have been removed from our SS-OOD experiments, following (Yang et al., 2022). Instead, we believe the CS-OOD setting is also significant for practical use. For instance, one may wish to distinguish real-world images from AI-generated ones (Zhang et al., 2024), or identify images that are severely contaminated due to environmental factors or sensor malfunctions (Baek et al., 2024). We present details of all datasets in Sec. B.1 and provide additional experimental results as well as details of reported results in Sec. C.

Table 1: Results on OpenOOD benchmark, averaged from 3 runs. The top results for each category are marked in bold, with the second-best result in underline. We include baseline results from Zhang et al. (2023), and reproduced the results for MC-Dropout (MCD). A dagger symbol † indicates direct access to training data or processes. Results are averaged separately for *near* / *far*-OOD sets.

| Method | CIFAR-10 | | CIFAR-100 | | ImageNet-200 | | ImageNet-1K | |
| --- | --- | --- | --- | --- | --- | --- | --- | --- |
| | FPR@95 ↓ | AUROC ↑ | FPR@95 ↓ | AUROC ↑ | FPR@95 ↓ | AUROC ↑ | FPR@95 ↓ | AUROC ↑ |
| MCD † | 53.54/31.43 | 87.68/91.00 | 54.73/59.08 | 80.42/77.58 | 55.25/35.48 | 83.30/90.20 | 65.68/51.45 | 76.02/85.23 |
| MDS † | 49.90/32.22 | 84.20/89.72 | 83.53/72.26 | 58.69/69.39 | 79.11/61.66 | 61.93/74.72 | 85.45/62.92 | 55.44/74.25 |
| ViM † | 44.84/25.05 | 88.68/**93.48** | 62.63/**50.74** | 74.98/**81.70** | 59.19/**27.20** | 78.68/91.26 | 71.35/24.67 | 72.08/92.68 |
| ODIN | 76.19/57.62 | 82.87/89.96 | 57.91/58.86 | 79.90/79.28 | 66.76/34.23 | 80.27/91.71 | 72.50/43.96 | 74.75/89.47 |
| EBO | 61.34/41.69 | 87.58/91.21 | 55.62/56.59 | 80.91/79.77 | 60.24/34.86 | 82.50/90.86 | 68.56/38.39 | 75.89/89.47 |
| MLS | 61.32/41.68 | 87.52/91.10 | 55.47/56.73 | 81.05/79.67 | 59.76/34.03 | 82.90/91.11 | **51.35**/63.60 | 76.46/89.57 |
| ASH | 86.78/79.03 | 75.27/78.49 | 65.71/59.20 | 78.20/80.58 | 64.89/27.29 | 82.38/**93.90** | 63.32/**19.49** | **78.17**/**95.74** |
| GEN | 53.67/47.03 | 88.20/91.35 | **54.42**/56.71 | **81.31**/79.68 | 55.20/32.10 | 83.68/91.36 | 65.32/35.61 | 76.85/89.76 |
| TULiP | **33.80**/24.43 | 89.67/92.55 | 55.07/58.17 | 81.29/79.63 | **54.51**/33.94 | **83.84**/91.03 | 64.96/48.01 | 77.52/88.03 |
| TULiP+GEN | 35.67/**23.51** | **90.04**/93.33 | 54.63/55.48 | 81.14/80.55 | 57.04/34.26 | 82.87/90.63 | 62.97/36.90 | 77.62/89.53 |

**Baseline Methods** We consider various baselines for comparison, including the MC-Dropout (MCD), post-hoc OOD methods without training data ODIN, EBO, MLS, ASH and GEN; and finally, MDS and ViM with access to training data. Please refer to Sec. 2 for a brief introduction.

**Semantic Shift OOD** We report the performance of TULiP on OpenOOD v1.5 benchmark (Zhang et al., 2023) in Table 1. Following their setup, we use the same pre-trained ResNet-18 (He et al., 2016) for CIFAR-10 & 100 (Krizhevsky, 2009) and ImageNet-200 (Zhang et al., 2023) ID datasets, and ResNet-50 for ImageNet-1K (Russakovsky et al., 2015). OOD data range across a collection of diverse image datasets (Cimpoi et al., 2014; Vaze et al., 2022; Van Horn et al., 2018; Bitterwolf et al., 2023; Le & Yang, 2015; Zhou et al., 2018; Kuznetsova et al., 2020), categorized into *near* and *far* OOD sets (Yang et al., 2022), where near is more similar to ID and therefore more difficult to distinguish. We also included a variant of TULiP+GEN as we substitute line 18 of Alg. 1 by GEN with $\gamma = 0.3$ and $M = 100$ to better demonstrate the effect of incorporating existing methods with TULiP. TULiP achieves remarkable performance in near-OOD settings with either top-1 or top-2 AUROC scores across all datasets. Indeed, as suggested by equation 8, better performance on near-ID scenarios is expected. On the far-OOD side, TULiP also performs consistently well. We note that methods significantly outperform TULiP on far-OOD either access the training dataset (ViM and MDS) or completely lack theoretical explanation (ASH). In ImageNet-1K (ResNet-50) AUROC, despite being outperformed by ASH, TULiP still outperforms other baselines by a large margin, with a slightly higher FPR. ASH is effective when the representation is redundant, as simplifying them does not significantly impact ID accuracy (Djurisic et al., 2023). ResNet-50, compared to ResNet-18 used otherwise, is more likely to have redundant representations due to its increased expressive power. In such cases, particularly in near-OOD scenarios, one may expect high performance for ASH when pruning parameters are appropriately tuned. On the other hand, TULiP demonstrates relatively consistent and high performance across all datasets. This indicates that properly evaluating uncertainty is fundamentally important, and our method achieves this goal to a considerable extent. Notably, ASH failed when using a different set of weights on ImageNet-1K (Appendix C.3). In contrast, TULiP, without access to training information, performs consistently well with theoretical foundations.

**Covariate Shift OOD** We test TULiP on the covariate shift setting with Blurred ImageNet, ImageNet-C (Hendrycks & Dietterich, 2019), ImageNet-R (Hendrycks et al., 2021) and ImageNet-ES (Baek et al., 2024) as OOD data. A description of these datasets can be found in Sec. B.1. For this experiment, the ImageNet validation set with blur is used for the hyper-parameter grid search. Table 2 reports the results. TULiP achieves top performance on ImageNet-C except for methods that require training data (MCD, MDS, ViM). This usually leads to longer evaluation time. For instance, ViM takes more than 30 minutes just to extract ID information using a recent GPU machine. In contrast, a typical full evaluation of TULiP on test split takes $3\times$ less time. On the other hand, ImageNet-R contains images that are less similar to ImageNet-1K (i.e. further from ID). When tuned on a near validation set like Blur-ImageNet, TULiP tends to favour near OOD by trading off the far ones. This is consistent with Table 1 and equation 8. Such phenomena are further demonstrated in Fig. 4.

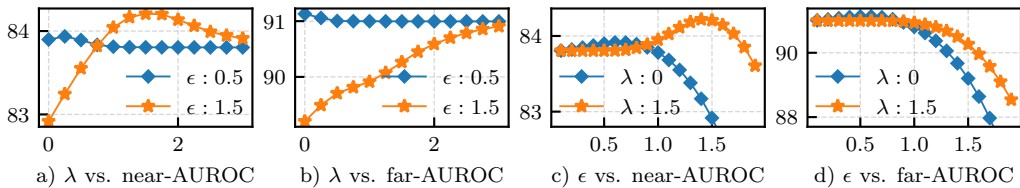

Figure 4: Results by varying $\epsilon$ and $\lambda$ on ImageNet-200 ID. The value of $\lambda$ in either the horizontal axis or legend should be read as, e.g., $\lambda = 1.5\sqrt{o}$.

**Network Architecture Choice** To verify TULiP across different network architectures, we conduct experiments with various networks on semantic-shift OOD with ImageNet-1K. The pre-trained models and weights are collected directly from *torchvision* (maintainers & contributors, 2016), and we only consider methods that work without additional modifications for compatibility. Results are shown in Fig. 3. TULiP relies on assumptions of the training process, which could be potentially violated by different training protocols and architectures. Nevertheless, TULiP still outperforms baseline methods consistently across the board, comparable to Table 1. Such results further suggest the effectiveness and versatility of TULiP.

Table 2: CS OOD results by averaging 3 runs. Results are in AUROC (higher is better).

| Method | Blur | ImNet-C | ImNet-R | ImNet-ES |
|---|---|---|---|---|
| MCD † | 69.90 | 77.06 | 80.52 | 79.98 |
| MDS † | 55.02 | 70.94 | 69.62 | 49.66 |
| ViM † | 73.88 | **83.93** | **87.92** | 82.54 |
| ODIN | 79.43 | 77.48 | 85.35 | 81.94 |
| EBO | 74.41 | 81.21 | 87.05 | 84.41 |
| MLS | 74.23 | 81.06 | 86.72 | 84.17 |
| ASH | 78.42 | 82.18 | 85.24 | 84.22 |
| TULiP | **85.34** | 82.91 | 82.07 | **85.91** |

**Ablation Study and Hyper-parameters** We conduct experiments on semantic-shifted ImageNet-200 to analyze the effect of hyper-parameters. Results are shown in Fig. 4, where we observe a trade-off in near and far OOD performance. It is also clear from the results that $\lambda$ and Lemma 3.2 boosts the performance and hyper-parameter stability (mainly to $\epsilon$) of TULiP. In practice, $\epsilon$ controls the overall strength of weight perturbation and, hence, the most important hyper-parameter of TULiP. Our method failed to achieve consistent performance across various datasets without layer-wise scaling, potentially due to its increased vulnerability to hyper-parameters and training setups. Please refer to Sec. C for more details.

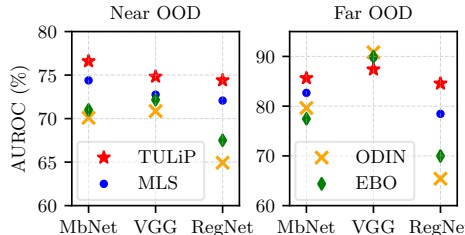

Figure 3: ImageNet-1K OpenOOD benchmarks on different network architectures: MobileNet V3 Large (MbNet) (Howard et al., 2019), VGG 16 (Simonyan & Zisserman, 2015), RegNet Y 16GF (Radosavovic et al., 2020).

## 6 CONCLUSION

In this study, we present TULiP, an uncertainty estimator for OOD detection. Our method is driven by the fluctuations under linearized training dynamics and excels in practical experiments. However, there are some limitations and future works remaining. Theoretically, our framework only considers functional perturbation. The perturbation on the NTK is also important (Kobayashi et al., 2022) and could be integrated into the estimator in the future. Furthermore, the layer-wise scaling scheme deserves more exploration as being discussed in Appendix C.2. Empirically, TULiP does not achieve state-of-the-art performance when the OOD data is far from ID (far-OOD). Such tradeoff in Fig. 4 hints at the inconsistency of best hyper-parameters for different setups. Future works may improve upon these aspects, covering a wider range of OOD data by examining the network parameters and refining weight perturbations. As shown in Appendix C.4, It is also beneficial to further develop TULiP for networks other than convolutional ones, such as transformers. In a broader aspect, exploring TULiP in other learning paradigms, such as Active Learning (Wang et al., 2022a) or Reinforcement Learning (Szepesvari, 2010) will be valuable.

## 7 REPRODUCIBILITY STATEMENT

We list our theoretical assumptions at the start of section 3.3, and all proofs thereafter in Appendix A. We provide a thorough overview of our experimental setup in section 5. A more detailed description of OOD configurations and additional results are presented in sections B and C of the Appendix, respectively. In the source codes provided in the supplementary materials, we include our implementation of the algorithm and the scripts to produce all visualizations. Additionally, we list the steps required to reproduce the OpenOOD results and provide a yaml file with all the hyper-parameters for the reported performance in this paper.

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

## A PROOFS

### A.1 BASIC NOTATIONS

For a network $f(\boldsymbol{x}) : \mathbb{R}^d \to \mathbb{R}^o$ maps inputs $\boldsymbol{x}$ of dimension $d$ to outputs $f(\boldsymbol{x})$ of dimension $o$, parameterized by $\boldsymbol{\theta}$ with $|\boldsymbol{\theta}|$ trainable parameters, the gradient / Jacobian matrix $\nabla_{\boldsymbol{\theta}} f(\boldsymbol{x})$ is a $o \times |\boldsymbol{\theta}|$ matrix.

The NTK $\Theta(\boldsymbol{z}, \boldsymbol{x}) := \nabla_{\boldsymbol{\theta}} f(\boldsymbol{z}) \nabla_{\boldsymbol{\theta}} f(\boldsymbol{x})^\top$ is a $o \times o$ matrix.

$\ell'(f_t(\boldsymbol{x}))$ is the gradient of loss function w.r.t. network output $f_t(\boldsymbol{x})$ at training time $t$. It is, for convenience, a $o \times 1$ column-vector.

The following lemma will be useful thereafter, which is an application of Hölder's inequality.

**Lemma A.1.** *Let $F : x \to \mathbb{R}^{m \times n}$, $g : x \to \mathbb{R}^n$. Consider 2-norms $\| \cdot \|$ (i.e., euclidean and its induced matrix 2-norm). For $p, q \in [1, \infty]$ that $\frac{1}{p} + \frac{1}{q} = 1$, we have*

$$\|\mathbb{E}_x[F(\boldsymbol{x})g(\boldsymbol{x})]\|$$
$$\leq \mathbb{E}_x[\|F(\boldsymbol{x})g(\boldsymbol{x})\|]$$
$$\leq \mathbb{E}_{\boldsymbol{x}}[\|F(\boldsymbol{x})\| \cdot \|g(\boldsymbol{x})\|]$$
$$\leq \mathbb{E}_{\boldsymbol{x}}[\|F(\boldsymbol{x})\|^p]^{1/p} \cdot \mathbb{E}_{\boldsymbol{x}}[\|g(\boldsymbol{x})\|^q]^{1/q}.$$

When $q = \infty$, we have $\mathbb{E}_{\boldsymbol{x}}[\|g(\boldsymbol{x})\|^q]^{1/q} := \sup_x \|g(\boldsymbol{x})\|$.

For convenience, given any random variable, vector or matrix $\mathbf{A}$ dependent of $\boldsymbol{x}$, we denote:

$$\|\mathbf{A}\|_X^{(q)} := \mathbb{E}_{\boldsymbol{x}}[\|\mathbf{A}\|^q]^{1/q}, \tag{16}$$

which by itself is a valid norm. We omit superscript $(q)$ if $q = 2$.

## A.2 ASSUMPTIONS

We recall the assumptions here, which are originally shown in Sec. 3. For network $f(\boldsymbol{x}, \boldsymbol{\theta})$, dataset $X$ with no parallel datapoints and a twice-differentiable loss function $\ell$, we assume the followings:

A1. (Boundedness) For $t \in [0, T]$, $f(\boldsymbol{x})$, $\nabla_{\boldsymbol{\theta}} f(\boldsymbol{x})$, $\ell$ and $\ell'$ stay bounded, uniformly on $\boldsymbol{x}$.

A2. (Smoothness) Gradient $\ell'$ of loss function $\ell$ is Lipschitz continuous: $\forall x \in X$; $\|\ell'(\hat{y}; y(\boldsymbol{x})) - \ell'(\hat{y}'; y(\boldsymbol{x}))\| \le L\|\hat{y} - \hat{y}'\|$.

A3. (Perturbation) The perturbation $\Delta f$ can be uniformly bounded by a constant $\alpha$, that is, for all $\boldsymbol{x}$ (not limited to the support of training data), i.e., $\forall x \in \mathbb{R}^d$; $\|\Delta f(\boldsymbol{x})\| \le \alpha$.

A4. (Convergence) Finally, for the original network trained via equation 3 and the perturbed network trained via equation 4, we assume *near-perfect convergence* on the training set $\boldsymbol{x}$ at termination time $t = T$, i.e., $\exists \beta \in \mathbb{R}, \forall x \in X$; $\|f_T(\boldsymbol{x}) - \hat{f}_T(\boldsymbol{x})\| \le \beta$.

## A.3 PROOF OF THEOREM 3.1

**Theorem A.2.** *(Theorem 3.1) Under assumptions A1-A4, for a network $f$ trained with equation 3 and a perturbed network $\hat{f}$ trained with equation 4, the perturbation applied at time $t_s = T - \Delta T$ bounded by $\alpha$, we have*

$$\|f_T(\boldsymbol{z}) - \hat{f}_T(\boldsymbol{z})\| \le \inf_{\boldsymbol{x} \in X} C\|\nabla_{\boldsymbol{\theta}} f(\boldsymbol{z}) - \nabla_{\boldsymbol{\theta}} f(\boldsymbol{x})\|_{\mathrm{F}} + 2\alpha + \beta, \tag{17}$$

*where $C = \frac{\alpha \eta \bar{\Theta}_X^{1/2}}{\lambda_{max}} \left(e^{(T-t_s)L\lambda_{max}} - 1\right)$, $\bar{\Theta}_X^{1/2} := \|\nabla_{\boldsymbol{\theta}} f(\boldsymbol{x})\|_X$ is the average gradient norm over training data, and $\lambda_{max} := \frac{1}{\sqrt{N}}\|\boldsymbol{G}\|$ for a generalized Gram matrix $G_{i,j} := \|\Theta(x_i, x_j)\|$ of dataset $X = \{x_1, x_2, \dots, x_N\}$.*

*Proof.* Let us first examine the fluctuations in the training set. From the Lipschitz continuity of $\ell'$,

$$\left\|\ell'(f_t(\boldsymbol{x})) - \ell'(\hat{f}_t(\boldsymbol{x}))\right\|_X \le L\|f(\boldsymbol{x}) - \hat{f}(\boldsymbol{x})\|_X. \tag{18}$$

Thus, by the linearized dynamics we have

$$\partial_t \left\|f_t(\boldsymbol{x}) - \hat{f}_t(\boldsymbol{x})\right\|_X$$

$$\le \left\|\partial_t \left(f_t(\boldsymbol{x}) - \hat{f}_t(\boldsymbol{x})\right)\right\|_X \tag{19}$$

$$= \left\|\mathbb{E}_{\boldsymbol{x}'}\left[\Theta(\boldsymbol{x}, \boldsymbol{x}')\left(\ell'(f_t(\boldsymbol{x}')) - \ell'(\hat{f}_t(\boldsymbol{x}'))\right)\right]\right\|_X$$

$$= \mathbb{E}_{\boldsymbol{x}}\left[\left\|\mathbb{E}_{\boldsymbol{x}'}\left[\Theta(\boldsymbol{x}, \boldsymbol{x}')\left(\ell'(f_t(\boldsymbol{x}')) - \ell'(\hat{f}_t(\boldsymbol{x}'))\right)\right]\right\|^2\right]^{1/2}$$

$$\le \mathbb{E}_{x}\left[\|\Theta(\boldsymbol{x}, \boldsymbol{x}')\|_X^2 \left\|\ell'(f_t(\boldsymbol{x}')) - \ell'(\hat{f}_t(\boldsymbol{x}'))\right\|_X^2\right]^{1/2}$$

$$\le \mathbb{E}_{x,x'}\left[\|\Theta(\boldsymbol{x}, \boldsymbol{x}')\|^2\right]^{1/2} \left\|\ell'(f_t(\boldsymbol{x})) - \ell'(\hat{f}_t(\boldsymbol{x}))\right\|_X$$

$$\le L\lambda_{max} \left\|f_t(\boldsymbol{x}) - \hat{f}_t(\boldsymbol{x})\right\|_X, \tag{20}$$

where in equation 19 we have used the triangle inequality to put $\partial_t$ inside the norm. $\lambda_{max}$ is defined as $\frac{1}{\sqrt{N}}\|\mathbf{G}\|$ for a generalized Gram-matrix $\mathbf{G}_{ij} := \|\Theta(x_i, x_j)\|$ of dataset $X = \{x_1, x_2, \dots, x_N\}$, measures the fitness (or alignment) of the kernel $\Theta$ w.r.t. the training data.

From equation 20, we can apply the Grönwall's inequality to obtain

$$\left\|f_t(\boldsymbol{x}) - \hat{f}_t(\boldsymbol{x})\right\|_X$$

$$\le \left\|f_{t_s}(\boldsymbol{x}) - \hat{f}_{t_s}(\boldsymbol{x})\right\|_X e^{(t-t_s)L\lambda_{max}}$$

$$\le \alpha e^{(t-t_s)L\lambda_{max}}. \tag{21}$$

We now prove Theorem 3.1 by generalizing equation 21 to given test data.

For a test point $z \in \mathbb{R}^d$, choose a pivot point $\boldsymbol{x}^* \in X$ from the training set. Then for the network function $f$ evaluated at $\boldsymbol{x}^*$ and $\boldsymbol{z}$, we have the followings:

$$
\begin{aligned}
&\left| \partial_t \left\| (f_t(\boldsymbol{z}) - f_t(\boldsymbol{x}^*)) - \left( \hat{f}_t(\boldsymbol{z}) - \hat{f}_t(\boldsymbol{x}^*) \right) \right\| \right| \\
&\leq \left\| \partial_t \left[ (f_t(\boldsymbol{z}) - f_t(\boldsymbol{x}^*)) - \left( \hat{f}_t(\boldsymbol{z}) - \hat{f}_t(\boldsymbol{x}^*) \right) \right] \right\| \\
&= \eta \left\| \mathbb{E}_{\boldsymbol{x}} \left[ (\Theta(\boldsymbol{z}, \boldsymbol{x}) - \Theta(\boldsymbol{x}^*, \boldsymbol{x})) \cdot \left( \ell'(f_t(\boldsymbol{x})) - \ell'(\hat{f}_t(\boldsymbol{x})) \right) \right] \right\|
\end{aligned}
\tag{22}
$$

Denote

$$
\left\| (f_t(\boldsymbol{z}) - f_t(\boldsymbol{x}^*)) - \left( \hat{f}_t(\boldsymbol{z}) - \hat{f}_t(\boldsymbol{x}^*) \right) \right\|
$$

as $\Delta f_t(\boldsymbol{z})$, and let $\Theta_{\boldsymbol{x}^*}^{\text{diff}}(\boldsymbol{z}, \boldsymbol{x}) := (\Theta(\boldsymbol{z}, \boldsymbol{x}) - \Theta(\boldsymbol{x}^*, \boldsymbol{x}))$. Integrate equation 22 with $t$, we have

$$
\begin{aligned}
&|\Delta f_T(\boldsymbol{z}) - \Delta f_{t_s}(\boldsymbol{z})| \\
&\leq \eta \int_{t_s}^T \left\| \mathbb{E}_{\boldsymbol{x}} \left[ \Theta_{\boldsymbol{x}^*}^{\text{diff}}(\boldsymbol{z}, \boldsymbol{x}) \left( \ell'(f_t(\boldsymbol{x})) - \ell'(\hat{f}_t(\boldsymbol{x})) \right) \right] \right\| dt \\
&\leq \eta \int_{t_s}^T \left\| \Theta_{\boldsymbol{x}^*}^{\text{diff}}(\boldsymbol{z}, \boldsymbol{x}) \right\|_X \left\| \ell'(f_t(\boldsymbol{x})) - \ell'(\hat{f}_t(\boldsymbol{x})) \right\|_X dt \\
&\leq \eta L \left\| \Theta_{\boldsymbol{x}^*}^{\text{diff}}(\boldsymbol{z}, \boldsymbol{x}) \right\|_X \int_{t_s}^T \left\| f_t(\boldsymbol{x}) - \hat{f}_t(\boldsymbol{x}) \right\|_X dt
\end{aligned}
\tag{23}
$$

We start with the term before the integral. To begin, rewrite it as:

$$
\begin{aligned}
&\| \Theta(\boldsymbol{z}, \boldsymbol{x}) - \Theta(\boldsymbol{x}^*, \boldsymbol{x}) \|_X \\
&= \left\| (\nabla_{\boldsymbol{\theta}} f(\boldsymbol{z}) - \nabla_{\boldsymbol{\theta}} f(\boldsymbol{x}^*)) \nabla_{\boldsymbol{\theta}} f(\boldsymbol{x})^\top \right\|_X \\
&\leq \mathbb{E}_x \left[ \| \nabla_{\boldsymbol{\theta}} f(\boldsymbol{z}) - \nabla_{\boldsymbol{\theta}} f(\boldsymbol{x}^*) \|^2 \cdot \| \nabla_{\boldsymbol{\theta}} f(\boldsymbol{x}) \|^2 \right]^{1/2} \\
&= \| \nabla_{\boldsymbol{\theta}} f(\boldsymbol{z}) - \nabla_{\boldsymbol{\theta}} f(\boldsymbol{x}^*) \| \cdot \| \nabla_{\boldsymbol{\theta}} f(\boldsymbol{x}) \|_X \\
&\leq \| \nabla_{\boldsymbol{\theta}} f(\boldsymbol{z}) - \nabla_{\boldsymbol{\theta}} f(\boldsymbol{x}^*) \|_{\mathrm{F}} \cdot \bar{\Theta}_X^{1/2},
\end{aligned}
\tag{24}
$$

where $\bar{\Theta}_X^{1/2} := \| \nabla_{\boldsymbol{\theta}} f(\boldsymbol{x}) \|_X$ is independent from $\boldsymbol{z}$.

**Remark.** *equation 24 used a computationally friendly Frobenius norm to bound the spectral norm in the line right above it. This is the main motivation to use a $\sqrt{o}$ scaling for hyper-parameter $\lambda$, as $\|\boldsymbol{A}\|_2 \leq \|\boldsymbol{A}\|_F \leq \sqrt{o}\|\boldsymbol{A}\|_2$ given $\boldsymbol{A}$ is full-rank $o \times |\boldsymbol{\theta}|$ and $o < |\boldsymbol{\theta}|$.*

Bring equation 24 and equation 21 back to equation 23 we have

$$
\begin{aligned}
&|\Delta f_T(\boldsymbol{z}) - \Delta f_{t_s}(\boldsymbol{z})| \\
&\leq \| \nabla_{\boldsymbol{\theta}} f(\boldsymbol{z}) - \nabla_{\boldsymbol{\theta}} f(\boldsymbol{x}^*) \|_{\mathrm{F}} \frac{\alpha \eta \bar{\Theta}_X^{1/2}}{\lambda_{max}} \left( e^{(T-t_s)L\lambda_{max}} - 1 \right).
\end{aligned}
\tag{25}
$$

Finally, we bound the difference between $f_T(\boldsymbol{z})$ and $\hat{f}_T(\boldsymbol{z})$ via the triangle inequality.

First, observe that from A3,

$$
\begin{aligned}
\Delta f_{t_s}(\boldsymbol{z}) &= \left\| (f_{t_s}(\boldsymbol{z}) - f_{t_s}(\boldsymbol{x}^*)) - \left( \hat{f}_{t_s}(\boldsymbol{z}) - \hat{f}_{t_s}(\boldsymbol{x}^*) \right) \right\| \\
&\leq \left\| f_{t_s}(\boldsymbol{z}) - \hat{f}_{t_s}(\boldsymbol{z}) \right\| + \left\| \hat{f}_{t_s}(\boldsymbol{x}^*) - f_{t_s}(\boldsymbol{x}^*) \right\| \\
&\leq 2\alpha,
\end{aligned}
\tag{26}
$$

so that

$$\Delta f_T(\boldsymbol{z}) \le |\Delta f_T(\boldsymbol{z}) - \Delta f_{t_s}(\boldsymbol{z})| + |\Delta f_{t_s}(\boldsymbol{z})|$$

$$\le \|\nabla_{\boldsymbol{\theta}} f(\boldsymbol{z}) - \nabla_{\boldsymbol{\theta}} f(\boldsymbol{x}^*)\|_{\mathrm{F}} \frac{\alpha \eta \bar{\Theta}_X^{1/2}}{\lambda_{max}} \left( e^{(T-t_s)L\lambda_{max}} - 1 \right) + 2\alpha. \tag{27}$$

Thus, given the convergence assumption $\|f_T(\boldsymbol{x}^*) - \hat{f}_T(\boldsymbol{x}^*)\| \le \beta$ (A4),

$$\left\| f_T(\boldsymbol{z}) - \hat{f}_T(\boldsymbol{z}) \right\|$$
$$= \left\| \left( f_T(\boldsymbol{z}) - f_T(\boldsymbol{x}^*) \right) - \left( \hat{f}_T(\boldsymbol{z}) - \hat{f}_T(\boldsymbol{x}^*) \right) - \left( \hat{f}_T(\boldsymbol{x}^*) - f_T(\boldsymbol{x}^*) \right) \right\|$$
$$\le \Delta f_T(\boldsymbol{z}) + \beta. \tag{28}$$

To proceed, recall that $\boldsymbol{x}^*$ is chosen arbitrarily. This concludes the proof. $\qquad\square$

**Note for equation 6:** Let $\boldsymbol{A} := \nabla_{\boldsymbol{\theta}} f(\boldsymbol{z}) - \nabla_{\boldsymbol{\theta}} f(\boldsymbol{x})$. Then, we have $\mathrm{Tr}\left(\boldsymbol{A}\boldsymbol{A}^\top\right) = \mathrm{Tr}\left(\Theta(\boldsymbol{z}, \boldsymbol{z}) + \Theta(\boldsymbol{x}, \boldsymbol{x}) - \Theta(\boldsymbol{z}, \boldsymbol{x}) - \Theta(\boldsymbol{x}, \boldsymbol{z})\right)$. Note that $\Theta(\boldsymbol{x}, \boldsymbol{z}) = \Theta(\boldsymbol{z}, \boldsymbol{x})^\top$ and therefore we may substitute them inside the trace. Thereafter, using $\|\boldsymbol{A}\|_{\mathrm{F}} = \left(\mathrm{Tr}(\boldsymbol{A}\boldsymbol{A}^\top)\right)^{1/2}$, we can obtain equation 6.

### A.4 PROOF OF LEMMA 3.2

We prove the lemma under the weakly lazy regime, i.e., we allow the weak dependency of $\Theta_t$ on $t$. Let us define $|\Theta_T(\boldsymbol{z}, \boldsymbol{x})|$ as the unique symmetric positive semi-definite solution of $|\Theta_T(\boldsymbol{z}, \boldsymbol{x})|^2 = \Theta_T(\boldsymbol{z}, \boldsymbol{x})^\top \Theta_T(\boldsymbol{z}, \boldsymbol{x})$, which is an extension of absolute values to matrices.

**Lemma A.3.** *(Extension of Lemma 3.2) We assume the lazy learning regime, i.e., there exists $\delta > 0$ such that $\sup_{\boldsymbol{x}, \boldsymbol{x}'} \||\Theta_T(\boldsymbol{x}, \boldsymbol{x}')| - |\Theta_t(\boldsymbol{x}, \boldsymbol{x}')|\| \le \delta$ holds for all $t_s \le t \le T$. Under assumption A1, with the model parameters $\boldsymbol{\theta}_T$ trained from $\boldsymbol{\theta}_{t_s}$ with equation 3 over the training set $\boldsymbol{x}$ and $t_s < T$, we have:*

$$\|\nabla_{\boldsymbol{\theta}} f_T(\boldsymbol{z})(\boldsymbol{\theta}_T - \boldsymbol{\theta}_{t_s})\| \le C(\mathrm{Tr}\mathbb{E}_{\boldsymbol{x}}[|\Theta_T(\boldsymbol{z}, \boldsymbol{x})|] + o\delta) + \sqrt{\delta}\|\boldsymbol{\theta}_T - \boldsymbol{\theta}_{t_s}\| \tag{29}$$

*where $C$ is a positive constant independent of $\boldsymbol{z}$.*

Lemma 3.2 is obtained by setting $\delta = 0$.

*Proof.* The mean value theorem for integrals guarantees that there exists $\tau \in [t_s, T]$ such that

$$\boldsymbol{\theta}_T - \boldsymbol{\theta}_{t_s} = -\int_{t_s}^T \eta \mathbb{E}_{\boldsymbol{x}} \left[ \nabla_{\boldsymbol{\theta}} f_\tau(\boldsymbol{x}) \ell'(f_\tau(\boldsymbol{x})) \right] dt. \tag{30}$$

Then, Hölder's inequality leads to

$$\|\nabla_{\boldsymbol{\theta}} f_\tau(\boldsymbol{z})(\boldsymbol{\theta}_T - \boldsymbol{\theta}_{t_s})\| = \left\| \mathbb{E}_{\boldsymbol{x}} \left[ \nabla_{\boldsymbol{\theta}} f_\tau(\boldsymbol{z}) \nabla_{\boldsymbol{\theta}} f_\tau(\boldsymbol{x})^\top \eta \int_{t_s}^T \ell'(f_\tau(\boldsymbol{x})) dt \right] \right\|$$

$$\le \|\Theta_\tau(\boldsymbol{z}, \boldsymbol{x})\|_X^{(1)} \cdot \underbrace{\left\| \eta \int_{t_s}^T \ell'(f_\tau(\boldsymbol{x})) dt \right\|_X^{(\infty)}}_{\text{independent of } \boldsymbol{z}}. \tag{31}$$

The lazy learning assumption leads that

$$\|\Theta_\tau(\boldsymbol{z}, \boldsymbol{x})\|_X^{(1)} \le \mathbb{E}_{\boldsymbol{x}} \left[ \mathrm{Tr} \left( \Theta_\tau(\boldsymbol{z}, \boldsymbol{x})^\top \Theta_\tau(\boldsymbol{z}, \boldsymbol{x}) \right)^{1/2} \right]$$
$$\le \mathbb{E}_{\boldsymbol{x}} \left[ \mathrm{Tr} \left( |\Theta_\tau(\boldsymbol{z}, \boldsymbol{x})| \right) \right]$$
$$\le \mathbb{E}_{\boldsymbol{x}} \left[ \mathrm{Tr} \left( |\Theta_T(\boldsymbol{z}, \boldsymbol{x})| \right) \right] + o\delta$$
$$= \mathrm{Tr} \left( \mathbb{E}_{\boldsymbol{x}} \left[ |\Theta_T(\boldsymbol{z}, \boldsymbol{x})| \right] \right) + o\delta.$$

Again the lazy learning assumption for $|\Theta_T(z, z)| = \Theta_T(z, z)$ ensures that

$$\|\nabla_{\boldsymbol{\theta}} f_\tau(\boldsymbol{z})(\boldsymbol{\theta}_T - \boldsymbol{\theta}_{t_s})\|^2 = (\boldsymbol{\theta}_T - \boldsymbol{\theta}_{t_s})^T \Theta_\tau(\boldsymbol{z}, \boldsymbol{z})(\boldsymbol{\theta}_T - \boldsymbol{\theta}_{t_s})$$
$$\geq (\boldsymbol{\theta}_T - \boldsymbol{\theta}_{t_s})^T (\Theta_T(\boldsymbol{z}, \boldsymbol{z}) - \delta \boldsymbol{I})(\boldsymbol{\theta}_T - \boldsymbol{\theta}_{t_s})$$
$$= \|\nabla_{\boldsymbol{\theta}} f_T(\boldsymbol{z})(\boldsymbol{\theta}_T - \boldsymbol{\theta}_{t_s})\|^2 - \delta \|\boldsymbol{\theta}_T - \boldsymbol{\theta}_{t_s}\|^2.$$

Hence, we have

$$\|\nabla_{\boldsymbol{\theta}} f_T(\boldsymbol{z})(\boldsymbol{\theta}_T - \boldsymbol{\theta}_{t_s})\| \leq \sqrt{\|\nabla_{\boldsymbol{\theta}} f_\tau(\boldsymbol{z})(\boldsymbol{\theta}_T - \boldsymbol{\theta}_{t_s})\|^2 + \delta \|\boldsymbol{\theta}_T - \boldsymbol{\theta}_{t_s}\|^2}$$
$$\leq \|\nabla_{\boldsymbol{\theta}} f_\tau(\boldsymbol{z})(\boldsymbol{\theta}_T - \boldsymbol{\theta}_{t_s})\| + \sqrt{\delta} \|\boldsymbol{\theta}_T - \boldsymbol{\theta}_{t_s}\|.$$

Substituting the above inequalities into equation 31, we obtain the conclusion of the lemma. $\square$

### A.5 PROOF OF PROPOSITION 4.1

**Proposition A.4.** *(Proposition 4.1) Suppose that $f^{emp}$ is $\gamma$-smooth w.r.t. $\boldsymbol{\theta}$, i.e.,*
$$\|\nabla_{\boldsymbol{\theta}} f^{emp}(\boldsymbol{z}; \boldsymbol{\theta}) - \nabla_{\boldsymbol{\theta}} f^{emp}(\boldsymbol{z}; \boldsymbol{\theta}')\|_{\mathrm{F}} \leq \gamma \|\boldsymbol{\theta} - \boldsymbol{\theta}'\|.$$
*Let $\mathbf{v}$ be a random variable such that $\mathbb{E}_{\mathbf{v}}[\mathbf{v}] = \mathbf{0}, \mathbb{E}_{\mathbf{v}}[\mathbf{v}\mathbf{v}^\top] = \epsilon^2 \boldsymbol{I}$ and $\mathbb{E}_{\mathbf{v}}[\|\mathbf{v}\|^k] \leq C_k \epsilon^k$ for $k = 3, 4$, where $C_k$ is a constant depending on $k$ and the dimension of $\mathbf{v}$. Then, under A1, it holds that*

$$\lim_{\epsilon \to 0} \frac{1}{\epsilon^2} \mathbb{E}_{\mathbf{v}} \left[ \|f^{emp}(\boldsymbol{z}; \boldsymbol{\theta}_T + \boldsymbol{\Gamma}\mathbf{v}) - f^{emp}(\boldsymbol{z}; \boldsymbol{\theta}_T)\|^2 \right] = \mathrm{Tr} \left( \nabla_{\boldsymbol{\theta}} f^{emp}(\boldsymbol{z}; \boldsymbol{\theta}_T) \boldsymbol{\Gamma}^2 \nabla_{\boldsymbol{\theta}} f^{emp}(\boldsymbol{z}; \boldsymbol{\theta}_T)^\top \right). \tag{32}$$

*Proof.* For each component $f_i^{emp}, i = 1, \ldots, o$, the mean value theorem leads that there exists $t_i \in [0, 1]$ such that

$$|f_i^{emp}(\boldsymbol{z}; \boldsymbol{\theta}_T + \boldsymbol{\Gamma}\mathbf{v}) - f_i^{emp}(\boldsymbol{z}; \boldsymbol{\theta}_T) - \nabla_{\boldsymbol{\theta}} f_i^{emp}(\boldsymbol{z}; \boldsymbol{\theta}_T)^\top \boldsymbol{\Gamma}\mathbf{v}|$$
$$= |\nabla_{\boldsymbol{\theta}} f_i^{emp}(\boldsymbol{z}; \boldsymbol{\theta}_T + t_i \boldsymbol{\Gamma}\mathbf{v})^\top \boldsymbol{\Gamma}\mathbf{v} - \nabla_{\boldsymbol{\theta}} f_i^{emp}(\boldsymbol{z}; \boldsymbol{\theta}_T)^\top \boldsymbol{\Gamma}\mathbf{v}|$$
$$\leq \gamma \|\boldsymbol{\Gamma}\|^2 \|\mathbf{v}\|^2.$$

For real numbers $a_i, b_i, i = 1, \ldots, o$, suppose $|a_i - b_i| \leq c$. Then, we have

$$\left| \sum_i a_i^2 - \sum_i b_i^2 \right| = \left| 2 \sum_i b_i(a_i - b_i) + \sum_i (a_i - b_i)^2 \right| \leq 2c \sum_i |b_i| + oc^2.$$

Using the above inequality, we obtain

$$\left| \mathbb{E}_{\mathbf{v}} \left[ \|f^{emp}(\boldsymbol{z}; \boldsymbol{\theta}_T + \boldsymbol{\Gamma}\mathbf{v}) - f^{emp}(\boldsymbol{z}; \boldsymbol{\theta}_T)\|^2 \right] - \mathbb{E}_{\mathbf{v}} \left[ \mathbf{v}^\top \boldsymbol{\Gamma} \nabla_{\boldsymbol{\theta}} f^{emp}(\boldsymbol{z}; \boldsymbol{\theta}_T)^\top \nabla_{\boldsymbol{\theta}} f^{emp}(\boldsymbol{z}; \boldsymbol{\theta}_T) \boldsymbol{\Gamma}\mathbf{v} \right] \right|$$

$$\leq 2\gamma \|\boldsymbol{\Gamma}\|^2 \mathbb{E}_{\mathbf{v}} [\sum_i |\mathbf{v}^\top \boldsymbol{\Gamma} \nabla_{\boldsymbol{\theta}} f_i^{emp}(\boldsymbol{z}; \boldsymbol{\theta}_T)| \|\mathbf{v}\|^2] + o\gamma^2 \|\boldsymbol{\Gamma}\|^4 \mathbb{E}_{\mathbf{v}}[\|\mathbf{v}\|^4]$$

$$\leq 2\sqrt{o}\gamma \|\boldsymbol{\Gamma}\|^3 \|\nabla_{\boldsymbol{\theta}} f^{emp}(\boldsymbol{z}; \boldsymbol{\theta}_T)\|_{\mathrm{F}} \mathbb{E}_{\mathbf{v}}[\|\mathbf{v}\|^3] + o\gamma^2 \|\boldsymbol{\Gamma}\|^4 \mathbb{E}_{\mathbf{v}}[\|\mathbf{v}\|^4]. \tag{33}$$

Note the cyclic trick for the trace ensures that

$$\mathbb{E}_{\mathbf{v}} \left[ \mathbf{v}^\top \boldsymbol{\Gamma} \nabla_{\boldsymbol{\theta}} f^{emp}(\boldsymbol{z}; \boldsymbol{\theta}_T)^\top \nabla_{\boldsymbol{\theta}} f^{emp}(\boldsymbol{z}; \boldsymbol{\theta}_T) \boldsymbol{\Gamma}\mathbf{v} \right]$$
$$= \mathbb{E}_{\mathbf{v}} \left[ \mathrm{Tr} \left( \boldsymbol{\Gamma} \nabla_{\boldsymbol{\theta}} f^{emp}(\boldsymbol{z}; \boldsymbol{\theta}_T)^\top \nabla_{\boldsymbol{\theta}} f^{emp}(\boldsymbol{z}; \boldsymbol{\theta}_T) \boldsymbol{\Gamma}\mathbf{v}\mathbf{v}^\top \right) \right]$$
$$= \mathrm{Tr} \left( \boldsymbol{\Gamma} \nabla_{\boldsymbol{\theta}} f^{emp}(\boldsymbol{z}; \boldsymbol{\theta}_T)^\top \nabla_{\boldsymbol{\theta}} f^{emp}(\boldsymbol{z}; \boldsymbol{\theta}_T) \boldsymbol{\Gamma} \mathbb{E}_{\mathbf{v}} \left[ \mathbf{v}\mathbf{v}^\top \right] \right)$$
$$= \mathrm{Tr} \left( \boldsymbol{\Gamma} \nabla_{\boldsymbol{\theta}} f^{emp}(\boldsymbol{z}; \boldsymbol{\theta}_T)^\top \nabla_{\boldsymbol{\theta}} f^{emp}(\boldsymbol{z}; \boldsymbol{\theta}_T) \boldsymbol{\Gamma} \cdot \epsilon^2 \boldsymbol{I} \right) \tag{34}$$
$$= \epsilon^2 \mathrm{Tr} \left( \nabla_{\boldsymbol{\theta}} f^{emp}(\boldsymbol{z}; \boldsymbol{\theta}_T) \boldsymbol{\Gamma}^2 \nabla_{\boldsymbol{\theta}} f^{emp}(\boldsymbol{z}; \boldsymbol{\theta}_T)^\top \right). \tag{35}$$

In equation 34 we applied the condition that $\mathbb{E}_{\mathbf{v}}[\mathbf{v}\mathbf{v}^\top] = \epsilon^2 \boldsymbol{I}$. We note that this is a slightly modified version of the well-known Hutchinson's Trace Estimator. We refer the readers to the existing analysis of such estimators (Avron & Toledo, 2011) for more details. As a result, we obtain

$$\left| \lim_{\epsilon \to 0} \frac{1}{\epsilon^2} \mathbb{E}_{\mathbf{v}} \left[ \|f^{emp}(\boldsymbol{z}; \boldsymbol{\theta}_T + \boldsymbol{\Gamma}\mathbf{v}) - f^{emp}(\boldsymbol{z}; \boldsymbol{\theta}_T)\|^2 \right] - \mathrm{Tr} \left( \nabla_{\boldsymbol{\theta}} f^{emp}(\boldsymbol{z}; \boldsymbol{\theta}_T) \boldsymbol{\Gamma}^2 \nabla_{\boldsymbol{\theta}} f^{emp}(\boldsymbol{z}; \boldsymbol{\theta}_T)^\top \right) \right|$$
$$\leq \lim_{\epsilon \to 0} 2\sqrt{o}\gamma \|\boldsymbol{\Gamma}\|^3 \|\nabla_{\boldsymbol{\theta}} f^{emp}(\boldsymbol{z}; \boldsymbol{\theta}_T)\|_{\mathrm{F}} C_3 \epsilon + o\gamma^2 \|\boldsymbol{\Gamma}\|^4 C_4 \epsilon^2$$
$$= 0.$$

The above equality means the conclusion of the proposition. $\square$

## A.6 ADDITIONAL DERIVATIONS FOR SECTION 4.3

Under the distribution of $\mathbf{v}$ we have

$$\mathbb{E}_{\mathbf{v}}[f^{\text{emp}}(\boldsymbol{z}; \boldsymbol{\theta}_T + \Gamma\mathbf{v})] = f^{\text{emp}}(\boldsymbol{z}; \boldsymbol{\theta}_T) + \underbrace{\mathbb{E}_{\mathbf{v}}[\nabla_\theta f^{\text{emp}}(\boldsymbol{z}; \boldsymbol{\theta}_T)^T \Gamma\mathbf{v}]}_{=0 \text{ from } \mathbb{E}_{\mathbf{v}}[\mathbf{v}]=0} + O(\mathbb{E}[\mathbf{v}^2])$$

$$= f^{\text{emp}}(\boldsymbol{z}; \boldsymbol{\theta}_T) + O(\epsilon^2),$$

which indicates that $\mathbb{E}_{\mathbf{v}}[f^{\text{emp}}(\boldsymbol{z}; \boldsymbol{\theta}_T + \Gamma\mathbf{v})] \approx f^{\text{emp}}(\boldsymbol{z}; \boldsymbol{\theta}_T)$ when $\epsilon$ is small.

We continue by the computation of $\text{TrVar}[\widetilde{f}^{\text{raw}}(\boldsymbol{z})]$:

$$\text{TrVar}_{\mathbf{v}}[\widetilde{f}^{\text{raw}}(\boldsymbol{z})] = \mathbb{E}_{\mathbf{v}}[\|f^{\text{emp}}(\boldsymbol{z}; \boldsymbol{\theta}_T + \Gamma\mathbf{v}) - \mathbb{E}_{\mathbf{v}}[f^{\text{emp}}(\boldsymbol{z}; \boldsymbol{\theta}_T + \Gamma\mathbf{v})]\|^2]$$

$$= \mathbb{E}_{\mathbf{v}}[\|f^{\text{emp}}(\boldsymbol{z}; \boldsymbol{\theta}_T + \Gamma\mathbf{v}) - f^{\text{emp}}(\boldsymbol{z}; \boldsymbol{\theta}_T) + O(\epsilon^2)\|^2]$$

$$= \mathbb{E}_{\mathbf{v}}[\|f^{\text{emp}}(\boldsymbol{z}; \boldsymbol{\theta}_T + \Gamma\mathbf{v}) - f^{\text{emp}}(\boldsymbol{z}; \boldsymbol{\theta}_T)\|^2] + O(\epsilon^4)$$

$$\approx \epsilon^2 \text{Tr}\Theta(\boldsymbol{z}, \boldsymbol{z}) + O(\epsilon^4).$$

Let $\widetilde{\Theta}_{\text{Tr}}(\boldsymbol{z}, \boldsymbol{z})$ be an approximation of $\epsilon^2 \text{Tr} \Theta(\boldsymbol{z}, \boldsymbol{z})$, which is being computed empirically in line 11 of Alg. 1.

Thus, $\gamma^2 \text{TrVar}_{\mathbf{v}}[\widetilde{f}^{\text{raw}}(\boldsymbol{z})]$ reads:

$$\gamma^2 \text{TrVar}_{\mathbf{v}}[\widetilde{f}^{\text{raw}}(\boldsymbol{z})] = \frac{[\widetilde{\Theta}_{\text{Tr}}(\boldsymbol{z}, \boldsymbol{z}) - \lambda D]_+}{\widetilde{\Theta}_{\text{Tr}}(\boldsymbol{z}, \boldsymbol{z})} \text{TrVar}_{\mathbf{v}}[\widetilde{f}^{\text{raw}}(\boldsymbol{z})]$$

$$\approx \frac{[\widetilde{\Theta}_{\text{Tr}}(\boldsymbol{z}, \boldsymbol{z}) - \lambda D]_+}{\widetilde{\Theta}_{\text{Tr}}(\boldsymbol{z}, \boldsymbol{z})} (\widetilde{\Theta}_{\text{Tr}}(\boldsymbol{z}, \boldsymbol{z}) + O(\epsilon^4))$$

$$\approx [\epsilon^2 \text{Tr} \Theta(\boldsymbol{z}, \boldsymbol{z}) - \lambda D]_+ + O(\epsilon^4)$$

where $[\,\cdot\,]_+$ denotes $\max(\cdot, 0)$.

For $D$, from approximation equation 13 we have

$$D = \left\| f^{emp}(\boldsymbol{z}; \boldsymbol{\theta}_T + \epsilon\delta\boldsymbol{\Gamma}(\boldsymbol{\theta}_T - \boldsymbol{\theta}_{t_s})) - f^{emp}(\boldsymbol{z}; \boldsymbol{\theta}_T) \right\|$$

$$\approx \epsilon\delta \left\| \nabla_{\boldsymbol{\theta}} f_T(\boldsymbol{z})(\boldsymbol{\theta}_T - \boldsymbol{\theta}_{t_s}) \right\|.$$

As a result, we have

$$\gamma^2 \text{TrVar}[\widetilde{f}^{\text{raw}}(\boldsymbol{z})] \approx \left[ \epsilon^2 \text{Tr} \Theta(\boldsymbol{z}, \boldsymbol{z}) - \epsilon\delta \left\| \nabla_{\boldsymbol{\theta}} f_T(\boldsymbol{z})(\boldsymbol{\theta}_T - \boldsymbol{\theta}_{t_s}) \right\| \right]_+.$$

Recall that equation 10 indicates that

$$\text{Tr}(\text{Var}_{\Delta f}[\hat{f}_T(\boldsymbol{z})]) \leq \mathbb{E}_{\Delta f}[\|\hat{f}_T(\boldsymbol{z}) - f_T(\boldsymbol{z})\|^2],$$

and Prop. 4.1 shows that

$$\|f_T(\boldsymbol{z}) - \hat{f}_T(\boldsymbol{z})\| \lesssim \left[ \text{Tr}(\Theta(\boldsymbol{z}, \boldsymbol{z}) + \underbrace{\mathbb{E}_x[\Theta(\boldsymbol{x}, \boldsymbol{x})]}_{\text{Independent of } \boldsymbol{z}}) - 2K \left\| \nabla_{\boldsymbol{\theta}} f_T(\boldsymbol{z})(\boldsymbol{\theta}_T - \boldsymbol{\theta}_{t_s}) \right\| \right]^{1/2}.$$

## A.7 PERTURB-THEN-TRAIN AND EQUATION 1

Jacot et al. (2018) consider neural networks in an infinite-width limit with specified initialization scheme, which we have referred as the lazy limit in Section 3. Under such limit, the linearized network equation 2 is justified as the empirical NTK (at initialization) converges to a specific deterministic kernel $\Theta$, where the distribution of a neural network $f(x; \boldsymbol{\theta})$'s initialization functional $f_{\text{Init}}(\boldsymbol{x})$ converges to a Gaussian Process (NNGP) (Lee et al., 2018a). In equation 2, it is equivalent to a deterministic (fixed) $\nabla_\theta f_{\text{True}}(x)|_{\theta=\theta^*}$ and a stochastic $f_{\text{Init}}$ following the NNGP.

Using the model defined in equation 2 and the training process described in equation 3, equation 1 effectively becomes:

$$\text{Var}_{f_{\text{Init}} \sim \mu_{\text{NNGP}}}[f_T(x; \theta | \text{Init} = f_{\text{Init}})], \tag{36}$$

where $f_T(x; \theta | \text{Init} = f_{\text{Init}})$ indicates a network trained via equation 3 by time $T$, with $f_{\text{Init}}$ as initialization.

When we set $t_s = 0$ (the initialization time), the perturbation $\Delta f$ will be applied to $f_{\text{Init}}$. Therefore, given a fixed initialization $f_0$ to perturb, Theorem 3.1 gives an upper-bound over a perturbation of the initialization functional:

$$\text{Var}_{\Delta f}[f_T(x; \theta | \text{Init} = f_0 + \Delta f)], \tag{37}$$

since $\hat{f}_T$ is supposed to be trained from initialization $f_0 + \Delta f$, we have $\hat{f}_T = f_T(x; \theta | \text{Init} = f_0 + \Delta f)$, hence the above.

Comparing it to equation 36, we see that the difference between them is the distribution of the initialization functional $f_{\text{Init}}$. In equation 36, $f_{\text{Init}}$ distributes according to the NNGP; while in equation 37, it is centered around $f_0$ with a stochastic perturbation $\Delta f$. Intuitively, by using theorem 3.1, we approximate the predictive variance trained from the NNGP prior with the predictive variance trained from a random perturbed initialization $f_0 + \Delta f$. Figure 2 visualizes such an approximation.

# B    DETAILS OF EXPERIMENTAL SETUP

## B.1    DATASET DESCRIPTIONS

An overview of all considered datasets is provided below. ID and OOD dataset setups are summarized in Table 3. Please refer to Zhang et al. (2023) for more details.

### B.1.1    ID DATASETS

**CIFAR-10**    The CIFAR-10 training set (Krizhevsky, 2009) consists 60000 $32 \times 32$ colored images, containing 10 classes of *airplane*, *automobile*, *bird*, *cat*, *deer*, *dog*, *frog*, *horse*, *ship* and *truck*. The test set originally contained 10000 images from the same classes, where we separated 1000 validation images and 9000 test images from the original test set following Zhang et al. (2023). The dataset and each split are even in classes.

**CIFAR-100**    CIFAR-100 (Krizhevsky, 2009) contains 60000 $32 \times 32$ images sampled from 100 classes, covering a wider range of images beyond CIFAR-10. Similar to CIFAR-10, 1000 images are taken out from the ID test set, forming a validation set.

**ImageNet-1K**    ImageNet-1K (Deng et al., 2009), also known as ILSVRC 2012, spans 1000 object classes and contains 1,281,167 training images, 50,000 validation images and 100,000 test images, each of size $224 \times 224$. In the OpenOOD setup, 45,000 validation images are used as ID test and 5,000 as ID validation.

**ImageNet-200**    ImageNet-200 (Zhang et al., 2023) is a 200-class subset of ImageNet-1K compiled in OpenOOD version 1.5, with 10,000 $224 \times 224$ validation images.

### B.1.2    SEMANTIC-SHIFT OOD DATASETS

**Tiny-ImageNet**    Tiny-ImageNet (Le & Yang, 2015) has 100,000 images divided up into 200 classes, each with 500 training images, 50 validating images, and 50 test images. Compared to ImageNet-200, every image in Tiny-ImageNet is downsized to a 64×64 coloured image.

**MNIST**    Modified National Institute of Standards and Technology database (Lecun et al., 1998) contains 60,000 training and 10,000 test images of handwritten digits. Each image is anti-aliased, normalized and centered to fit into a 28x28 pixel bounding box.

**SVHN**    Street View House Number (Netzer et al., 2011) dataset contains house numbers that are captured on Google Street View, consisting of 73257 digits for training, and 26032 digits for testing. In our setup, we used the MNIST-like 32-by-32 format, centered around a single character.

**Textures** Describable Textures Dataset (Cimpoi et al., 2014) is a set of 47 categories of textures, collected from Google and Flickr via relevant search queries. It has 5640 images, 120 images for each category, where the sizes range between 300x300 and 640x640.

**Places365** Places365 (Zhou et al., 2018) is a scene recognition dataset. The standard version is composed of 1.8 million train and 36000 validation images from 365 scene classes.

**NINCO** No ImageNet Class Objects (Bitterwolf et al., 2023) consists of 5879 samples from 64 OOD classes. These OOD classes were selected to have no categorical overlap with any classes of ImageNet-1K. Each sample was inspected individually by the authors to not contain ID objects.

**SSB-Hard** Semantic Shift Benchmark-Hard (Vaze et al., 2022) split contains 49,000 images across 980 categories of ImageNet-21K () that has a short total semantic distance.

**iNaturalist** The iNaturalist dataset (Van Horn et al., 2018) has 579,184 training and 95,986 validation images from 5,089 different species of plants and animals.

**OpenImage-O** OpenImage-O (Kuznetsova et al., 2020) is image-by-image filtered from the test set of OpenImage-V3, which has been collected from Flickr without a predefined list of class names or tags. In the OpenOOD setup, 1,763 images are picked out as validation OOD.

### B.1.3 COVARIATE-SHIFT OOD DATASETS

**Blur-ImageNet** This blurred ImageNet dataset contains ImageNet images with a Gaussian blur of $\sigma = 2$. The same splits are used as in the above description in the ImageNet-1K section.

**ImageNet-C** ImageNet-C (Hendrycks & Dietterich, 2019) has 15 synthetic corruption types (such as noise, blur, pixelate) on the standard ImageNet-1K, each with 5 severities. In OpenOOD, 10,000 images are randomly sampled uniformly across the 75 combinations to form the test set.

**ImageNet-R** ImageNet-R (Hendrycks et al., 2021) contains 30,000 images of different renditions of 200 ImageNet classes, such as art, graphics, patterns, toys, and video games.

**ImageNet-ES** ImageNet-ES (Baek et al., 2024) consists of 202,000 photos of images from Tiny-ImageNet. Each image is displayed on screen with high fidelity and photographed in a controlled environment with different parameter settings. We only used the 64,000 photos in the test set.

### B.2 HYPER-PARAMETERS

During our preliminary experiments, we found that the setup given in the main text ($\lambda = \sqrt{o}$, $\epsilon = 2$, $\delta = 2$) works consistently well across datasets. In this preliminary stage, we have only considered smaller datasets such as MNIST, CIFAR-10, SVHN, etc., as well as ImageNet-blur. During future developments, larger $\delta$ and $\lambda$ show better performance on large-scale datasets, as we have included them in the hyper-parameter searching range of TULiP, whenever a validation set is available. We further extended the range of $\epsilon$ to improve the performance of TULiP for different network architectures and training setups.

In practice, when handling hyper-parameters, we found it beneficial to first search an optimal value for $\epsilon$, the most important parameter of TULiP as pointed out in Sec. 5, while fixing $\lambda$ and $\delta$ as suggested. It controls the overall strength of weight perturbation and may depends on network architecture and training scheme. If one's computational resource allows for further exploration, optimal values of $\lambda$ and $\delta$ can be searched for better performance. If a validation set is not available, one may either use the suggested value or investigate network outputs after weight perturbation. When the network output become senseless after perturbation (e.g., a prediction close to random-guessing), it often indicates that $\epsilon$ is too large.

### B.2.1 GRID SEARCH

Table 4 lists the hyper-parameter search range for all considered methods on the validation set.

Table 3: ID, OOD and OOD-val dataset setups.

| ID Dataset | near-OOD | far-OOD | near/far-OOD Validation | Cov-Shift OOD |
|---|---|---|---|---|
| CIFAR-10 | CIFAR-100 Tiny-ImageNet | MNIST SVHN Textures Places365 | Tiny-ImageNet | |
| CIFAR-100 | CIFAR-10 Tiny-ImageNet | MNIST SVHN Textures Places365 | Tiny-ImageNet | |
| ImageNet-1K ImageNet-200 | SSB-Hard NINCO | iNaturalist Textures OpenImage-O | OpenImage-O | Blur-ImageNet ImageNet-C ImageNet-R ImageNet-ES |

Table 4: Hyper-parameter (available at evaluation time) search ranges.

| Method | Hyper-parameters |
|---|---|
| MC-Dropout | N/A |
| MDS | N/A |
| MLS | N/A |
| EBO | Temperature: $\{1\}$ |
| ViM | Dimension: $\{256, 1000\}$ |
| ASH | Percentile: $\{65, 70, 75, 80, 85, 90, 95\}$ |
| ODIN | Temperature: $\{1, 10, 100, 1000\}$ Noise: $\{0.0014, 0.0028\}$ |
| TULiP | $\delta$: $\{2, 5, 8\}$, $\lambda$: $\{1, 3\} \cdot \sqrt{o}$ $\epsilon$: $\{0.1, 0.5, 1.5, 2.0\}$ |

### B.2.2 $\sqrt{o}$ SCALING OF $\lambda$

In practice, when the number of network output dimensions $o$ varies, we found that a $\sqrt{o}$ scaling of $\lambda$ works more consistently. This might be due to our choice of using the computational-friendly Frobenius norm in Theorem 3.1 instead of a tighter spectrum norm. It is further explained in the proof listed in Sec. A.3.

### B.2.3 HARDWARE

Each of our experiments is conducted on a single-node machine using an NVIDIA A6000 GPU.

## C ADDITIONAL EXPERIMENT RESULTS

### C.1 DETAILED RESULTS

Following Zhang et al. (2023), we used the provided pre-trained weights from 3 training runs for the results reported for CIFAR-10, CIFAR-100 and ImageNet-200 ID datasets. For ImageNet-1K ID, 3 evaluation (OOD) runs on a single training run is reported. Results for each individual run, as well as each individual dataset, are listed in Table 5.

### C.2 EFFECT OF LAYER-WISE SCALING

Layer-wise scaling is an essential component of TULiP. In this experiment, we conduct semantic-shift OOD detection on ImageNet-1K without layer-wise scaling. In particular, for networks with $L$ layers

Table 5: Detailed breakdown of TULiP's Semantic-shift OOD performance on individual datasets.

| ID dataset | OOD dataset | Run 1 FPR95↓ | Run 1 AUROC↑ | Run 2 FPR95↓ | Run 2 AUROC↑ | Run 3 FPR95↓ | Run 3 AUROC↑ |
|---|---|---|---|---|---|---|---|
| CIFAR-10 | CIFAR-100 | 36.11 | 88.86 | 37.56 | 88.56 | 36.52 | 88.81 |
| | Tiny-ImageNet | 30.67 | 90.75 | 31.42 | 90.23 | 30.53 | 90.83 |
| | Near OOD | 33.39 | 89.81 | 34.49 | 89.39 | 33.53 | 89.82 |
| | MNIST | 13.83 | 96.80 | 14.76 | 96.44 | 17.07 | 95.16 |
| | SVHN | 23.02 | 91.75 | 21.30 | 93.06 | 20.59 | 93.17 |
| | Textures | 30.96 | 89.98 | 30.49 | 90.60 | 28.12 | 91.44 |
| | Places365 | 29.31 | 91.49 | 31.11 | 90.69 | 32.68 | 90.09 |
| | Far OOD | 24.28 | 92.50 | 24.41 | 92.70 | 24.61 | 92.46 |
| CIFAR-100 | CIFAR-10 | 60.93 | 79.24 | 58.80 | 79.35 | 60.92 | 78.90 |
| | Tiny-ImageNet | 48.96 | 83.66 | 50.01 | 83.50 | 50.80 | 83.08 |
| | Near OOD | 54.94 | 81.45 | 54.41 | 81.43 | 55.86 | 80.99 |
| | MNIST | 57.83 | 79.36 | 47.88 | 84.45 | 53.82 | 80.74 |
| | SVHN | 58.82 | 79.49 | 58.68 | 80.42 | 63.07 | 77.94 |
| | Textures | 61.48 | 78.65 | 60.67 | 78.10 | 64.20 | 76.82 |
| | Places365 | 56.41 | 80.21 | 57.51 | 79.66 | 57.68 | 79.69 |
| | Far OOD | 58.64 | 79.43 | 56.18 | 80.66 | 59.69 | 78.80 |
| ImageNet-200 | SSB-hard | 65.87 | 80.86 | 66.20 | 80.89 | 65.39 | 80.91 |
| | NINCO | 43.94 | 86.68 | 42.48 | 86.85 | 43.14 | 86.84 |
| | Near OOD | 54.91 | 83.77 | 54.34 | 83.87 | 54.27 | 83.88 |
| | iNaturalist | 22.52 | 93.80 | 22.94 | 93.45 | 24.80 | 93.21 |
| | Textures | 45.00 | 89.53 | 43.20 | 89.80 | 44.02 | 89.67 |
| | OpenImage-O | 34.64 | 89.98 | 33.99 | 89.94 | 34.34 | 89.92 |
| | Far OOD | 34.06 | 91.10 | 33.38 | 91.06 | 34.39 | 90.93 |
| ImageNet-1K | SSB-hard | 74.09 | 73.16 | 73.99 | 73.37 | 74.12 | 73.14 |
| | NINCO | 55.92 | 81.83 | 55.75 | 81.82 | 55.89 | 81.81 |
| | Near OOD | 65.00 | 77.50 | 64.87 | 77.59 | 65.01 | 77.48 |
| | iNaturalist | 37.89 | 91.01 | 38.08 | 90.98 | 37.94 | 91.01 |
| | Textures | 59.02 | 85.34 | 59.15 | 85.29 | 59.01 | 85.34 |
| | OpenImage-O | 47.10 | 87.77 | 46.87 | 87.76 | 47.08 | 87.76 |
| | Far OOD | 48.00 | 88.04 | 48.03 | 88.01 | 48.01 | 88.04 |

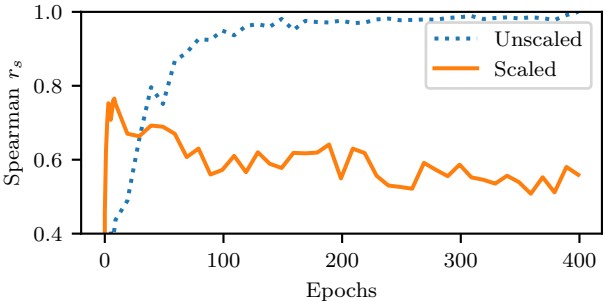

Figure 5: Visualization of the effect of layer-wise scaling (solid, orange) vs. without layer-wise scaling (dotted, blue). The vertical axis indicates the Spearman rank correlation between the direct calculation of empirical NTK in training $\mathrm{Tr}\left(\nabla_{\boldsymbol{\theta}} f_t^{emp}(\boldsymbol{x})\nabla_{\boldsymbol{\theta}} f_t^{emp}(\boldsymbol{x}')^{\top}\right)$ and scaled NTK after training $\mathrm{Tr}\left(\nabla_{\boldsymbol{\theta}} f_T^{emp}(\boldsymbol{x})\boldsymbol{\Gamma}^2\nabla_{\boldsymbol{\theta}} f_T^{emp}(\boldsymbol{x}')^{\top}\right)$, for $T=$ epoch 400 and $t$ spanning the horizontal axis. The network is a ResNet-18 variant trained on CIFAR-10 for 400 epochs with SGD momentum 0.9, and $\boldsymbol{x}, \boldsymbol{x}'$ are sampled from the training set for 4096 pairs. For solid orange curve $\boldsymbol{\Gamma} = (1/\sqrt{|\boldsymbol{\theta}_l|}) \cdot \boldsymbol{I}$ (scaled) and for dotted blue curve $\boldsymbol{\Gamma} \propto \boldsymbol{I}$ (unscaled). It indicates that the proposed layer-wise scaling scheme helps recovering an earlier network state, as the scaled NTK is more similar to the early empirical NTKs.

Table 6: OOD detection results (AUROC ↑) for TULiP without layer-wise scaling (w/o LW). TULiP results are copied from Table 1.

| Method | CIFAR-10 | CIFAR-100 | ImNet-200 | ImNet-1K | ImNet-Blur | ImNet-C | ImNet-R |
|---|---|---|---|---|---|---|---|
| TULiP | 89.67/92.55 | **81.29/79.63** | **83.84/91.03** | **77.52/88.03** | **85.54** | 82.91 | 82.07 |
| w/o LW | **90.68/92.81** | 80.47/77.90 | 81.54/86.75 | 76.32/84.99 | 76.86 | **83.76** | **85.30** |

and parameters of layer $l$ denoted as $\boldsymbol{\theta}_l$, the scaling matrix $\boldsymbol{\Gamma}$ has been set to $(L^{-1}\sum_l 1/\sqrt{|\boldsymbol{\theta}_l|}) \cdot \boldsymbol{I}$, i.e., an averaged scaling $\boldsymbol{\Gamma} \propto \boldsymbol{I}$ is used for the entire network, identical across layers, effectively disables layer-wise scaling while maintaining a similar magnitude for perturbations.

Table 6 compares the results of this experiment and the one reported in Table 1 and 2. It shows the effect of layer-wise scaling on TULiP. Intuitively, it helps to find an NTK that is more representative of the training process, reducing the gap between the linearized training trajectory and the true training trajectory. Such effect is further demonstrated in Figure 5. From an empirical aspect, our scaling is often approximately proportional to the magnitude of individual parameters within the layer (cf. Fig. 1 a) of the main paper). In this sense, a larger perturbation may significantly interfere with the network performance, thus producing unpredictable results. Our Layer-wise scaling scheme also reduces such vulnerability by applying smaller perturbations to layers with smaller weights. Nevertheless, TULiP without layer-wise scaling still outperforms TULiP in some datasets, suggesting future work for an in-depth analysis.

### C.3 V2 WEIGHTS ON IMAGENET-1K

Recently, researchers have been finding the possibilities to extend the performance of existing models such as ResNet-50 on various datasets. For example, *torchvision* (maintainers & contributors, 2016) released a new version (V2) of a ResNet-50 trained on ImageNet-1K with recent advances in practical NN training, increasing the Top-1 accuracy by 4.7% (Vryniotis, 2021). The previous version (V1) is used in OpenOOD v1.5 for ImageNet-1K ID and ResNet-50 backbone (Zhang et al., 2023).

Somewhat surprisingly, the performance of all OOD detectors are severely undermined when using the new V2 weights. We summarize our empirical findings in Table 7. Upon further inspection, we have empirically found that in general, V2 weight is larger than V1, especially the $\gamma$, $\beta$ parameters in BatchNorm layers (Ioffe & Szegedy, 2015) have been increased around 10×. This results in a significantly larger $\|\nabla_{\boldsymbol{\theta}} f(\boldsymbol{x})\|$. As a consequence, the network is more vulnerable to weight

Table 7: ImageNet-1K OOD AUROC score using the V2 weights from *torchvision*.

|  | MLS | ODIN | ViM | ASH | TULiP (Ours) |
|---|---|---|---|---|---|
| SSB-Hard | 65.53 | 69.03 | 57.93 | 40.52 | 67.57 |
| NINCO | 72.96 | 70.81 | 72.42 | 31.39 | 74.90 |
| Near OOD | 69.24 | 69.92 | 65.17 | 35.96 | **71.23** |
| iNaturalist | 80.34 | 67.19 | 92.32 | 28.64 | 81.62 |
| Textures | 71.42 | 62.95 | 95.04 | 34.97 | 70.20 |
| OpenImage-O | 77.66 | 68.22 | 89.89 | 26.13 | 79.02 |
| Far OOD | 76.47 | 66.12 | **92.42** | 29.91 | 76.95 |

Table 8: Semantic Shift-OOD on ImageNet-1K with ViT-B-16 model. Baseline results cited from Zhang et al. (2023).

| | ImageNet-1K | |
|---|---|---|
| Method | FPR@95 ↓ | AUROC ↑ |
| ViM† | 73.73/29.18 | 77.03/92.84 |
| MDS† | 66.12/29.97 | 79.04/92.60 |
| EBO | 93.19/85.35 | 62.41/78.98 |
| MLS | 92.25/79.23 | 68.30/83.54 |
| ASH | 94.43/96.77 | 53.21/51.56 |
| GEN | 70.78/32.23 | 76.30/91.35 |
| TULiP | 84.73/52.23 | 73.63/87.98 |

perturbations and thus favors a much smaller $\epsilon$. In particular, the results in Table 7 were produced with a perturb power of $\epsilon = 0.1$ (chosen with respect to the validation set). Such phenomena further demonstrate the significant effect of $\epsilon$ to TULiP. It hints at an important future research direction that aims to tackle such sensitivity.

### C.4 POST-HOC METHODS AND VISION TRANSFORMERS (VIT)

In this subsection, following Zhang et al. (2023), we report the results of a direct implementation of TULiP on ViT-B-16 (Dosovitskiy et al., 2021) in Table 8. The same setup as in Semantic-Shift OOD experiments has been used except for the network architecture. Thanks to their superior performance, transformer-based models have become one of the mainstream models in the vision literature ever since they have been adopted to the field. Interestingly, as shown in Table 8, almost all post-hoc methods without training data access degrade their performance compared to their convolution-based performance in Table 1, despite the increased expression power of ViT. Those results suggest that one may need specific tuning to make post-hoc methods perform better on transformer models. For TULiP, one of the specific tunings could be to introduce architectural knowledge of transformers, in order to obtain a more accurate approximation.

### C.5 ADDITIONAL EXPERIMENT FOR OUTLIER REJECTION

Following established protocols (Krishnan & Tickoo, 2020; Thiagarajan et al., 2022), we conduct OOD detection experiments with ImageNet-1K as inliers and images with Gaussian blur of intensity 5 from ImageNet-C (Hendrycks & Dietterich, 2019) as outliers. For TULiP, we used the hyper-parameters suggested in Sec. 5 as there are no validation sets in this experiment.

**Additional Baselines** Thiagarajan et al. (2022) proposed $\Delta$-UQ, a method for Uncertainty Quantification that utilizes training data as anchors to create network ensembles, where each instance uses a different anchor. SVI (Blundell et al., 2015b), stands for stochastic variational inference, is a Bayesian UQ method utilizing variational inference. Temperature Scaling (Guo et al., 2017b) is a simple post-hoc UQ method that scales the logits before the softmax layer to estimate prediction uncertainty.

Table 9: Outlier rejection results with ImageNet-C Gaussian Blur intensity 5. Baseline results are copied from (Thiagarajan et al., 2022). † represents training data access.

| Method | | AUROC ↑ | AUPR-in/out ↑ |
|---|---|---|---|
| (Lakshminarayanan et al., 2017) | Deep Ensembles † | 95.49 | 95.31 / 95.64 |
| (Gal & Ghahramani, 2016) | MC Dropout † | 96.38 | 96.16 / 96.67 |
| (Blundell et al., 2015b) | SVI † | 96.40 | 95.97 / 96.83 |
| (Thiagarajan et al., 2022) | $\Delta$-UQ † | **97.49** | **97.56 / 97.47** |
| (He et al., 2016) | ResNet-50 | 93.36 | 92.82 / 93.71 |
| (Guo et al., 2017b) | Temperature Scaling | 93.71 | 93.21 / 94.01 |
| (ours) | TULiP | **96.40** | **96.58 / 96.32** |

Table 10: Wall-clock time of our SS-OOD experiments, contains a serial sequence of inference on ID (top row) and all corresponding OOD datasets (near and far).

| Method | Forward passes | CIFAR-10 | ImageNet-200 | ImageNet-1K |
|---|---|---|---|---|
| EBO | 1 | 44.32s | 112.37s | 3m 12.60s |
| TULiP | $\mathcal{O}(M)$, $M = 10$ | 96.30s (2.17x) | 190.41s (1.69x) | 10m 59.24s (3.42x) |

In Table 9, we report our results with baseline results copied from Thiagarajan et al. (2022). It is worth noting that in this experiment, TULiP, despite being a post-hoc method, outperforms many UQ methods that would require significantly more computational resources (e.g., Deep Ensembles, MC Dropout, etc.). It is on par with SVI and being outperformed by $\Delta$-UQ, which is a much heavier method that relies on network architecture modifications before training (thus requires training the network from stretch) and domain-specific data augmentations.

## C.6 TIME COMPLEXITY OF TULiP

As listed in Algorithm 1, TULiP requires $\mathcal{O}(M)$ forward passes to evaluate a minibatch of test data. Compared to single-pass methods, such limitation renders TULiP ineffective despite its performance as shown in Sec. 5, since forwarding a network could be expensive as networks grow in size. Nevertheless, TULiP is not $\mathcal{O}(M)$ times slower than single-pass methods as forward evaluation is not the sole bottleneck of inferencing. Table 10 compares the wall-clock inference time of TULiP and EBO (a single-pass method) in our SS-OOD setting.

