# OpenReview forum: "TULiP: Test-time Uncertainty Estimation via Linearization and Weight Perturbation"
_ICLR.cc/2025/Conference — Submitted to ICLR 2025_

### Official Review · Reviewer_enPE · 2024-10-28

**Soundness:** 2
**Presentation:** 2
**Contribution:** 2
**Rating:** 5
**Confidence:** 4

**Summary:**

This work introduces TULIP (Test-time Uncertainty by Linearized fluctuations via weight Perturbation), a post-hoc out-of-distribution (OoD) score that leverages the epistemic uncertainty of a trained network. Grounded in the theoretical framework of linearized training dynamics, TULIP demonstrates effectiveness in detecting both semantic-shift and covariate-shift OoD scenarios.

**Strengths:**

- This work proposes an uncertainty-based score to detect both semantic-shift OOD and covariate-shift OOD without accessing the training data.
- The derivation of the proposed bound for ||f_T(x)-\hat{f}_T(x)|| and its upper bound are written down thoroughly.

**Weaknesses:**

- Line 018: Could the authors clarify what is meant by “other problem settings”?

- Connection Between Concepts (Line72-73): The relationship between semantic shift and covariate shift in Out-of-Distribution (OOD) detection and epistemic uncertainty is not clearly motivated [1].  Could the authors provide further elaboration on this connection?
- Post-hoc OOD Detectors: The related work section appears somewhat outdated and incomplete. A notable aspect of TULiP is its ability to perform OOD detection without access to the training data, which aligns it with post-hoc detection methods. Additionally, TuLiP addresses both semantic-shift and covariate-shift OOD detection. Therefore, the related work should be expanded to include recent studies on post-hoc detectors for semantic-shift OOD [2] and covariate-shift OOD [3], respectively.

- Clarity of the Paper: The overall clarity of the paper could be improved. For instance, the section on the theoretical framework (particularly sections 3.1 and 3.2) could be condensed, as it does not represent a primary contribution of this work.

- The discussion on the bound for the ||f_T(x)-\hat{f}_T(x)|| and its upper bound is appreciated. However, the implementation section (Section 4) is somewhat unclear. For example, the “early checkpoint” in Algorithm 1 is set to zero by default (Line 376), though it is marked as optional. Additionally, the OOD score is based on Shannon entropy, which has been explored previously in [2]. The primary difference appears to be that [2] directly utilizes the predictive distribution from a trained network, whereas TULiP relies on perturbed predictions.

- Performance on ImageNet-1k: When evaluating OOD detection on large-scale benchmarks using ImageNet-1k as the in-distribution data, the method’s performance in terms of AUROC and FPR95 is worse than ASH (see Table 1). This discrepancy raises questions about the claim in the abstract regarding TULiP achieving state-of-the-art (SOTA) performance. Additionally, a discussion on why performance degrades with ImageNet-1k as the in-distribution data would be helpful.

- Optional Suggestion for Table 1: For improved readability, Table 1 could be divided into two sections: one for methods that do not require training data access, and another for those that do.

- Additional Architectures: Testing on alternative architectures, such as BiT [4] and ViT [5], would further demonstrate the robustness and effectiveness of the proposed approach.

- Results on Table 2: It seems inappropriate to compare methods designed specifically for semantic-shift OOD detection, as the current task focuses on covariate-shift OOD detection.  Additionally, could the authors clarify the metric reported in Table 2?  The purpose of TULiP is somewhat ambiguous—whether it aims to detect covariate-shift OOD samples or to demonstrate robustness against them. If the latter is the case, a fairer comparison would be with methods developed explicitly for covariate-shift OOD samples, such as [3].

 References
- [1] Aleatoric and epistemic uncertainty in machine learning: an introduction to concepts and methods. Machine Learning, 110(3):457–506, 2021.
- [2] GEN: Pushing the Limits of Softmax-Based Out-of-Distribution Detection. In CVPR, 2023.
- [3] Semantically Coherent Out-of-Distribution Detection. In ICCV, 2023.
- [4] Big transfer (bit): General visual representation learning. In ECCV, 2020.
- [5] An image is worth 16x16 words: Transformers for image recognition at scale. In ICLR, 2021.

**Questions:**

See weakness

---

> ### Author Response · Authors · 2024-11-22
>
> We are grateful for your thorough and constructive feedback on our manuscript.
>
> We uploaded a revision of our manuscript. In the following, unless otherwise mentioned, we refer to sections and figures/tables in our original manuscript.
>
> We are further revising the paper regarding your legitimate comments, especially on lines 18, 72-73, related works, tables, clarity in Sections 3 and 4, and parts related to early checkpoints.
>
> **part of W5: Regards to GEN [2]**
>
> Thank you for recommending the study [2] as our related work. We will revise our manuscript by including it in the related work section. Here, let us remark on some differences as follows: To be precise, our work differs from GEN [2], since we are focusing on an approximation of the posterior samples rather than Shannon Entropy itself, and perturbation plays a central role in our method. In Algorithm 1, Shannon Entropy is employed as it is a common choice to estimate predictive uncertainty from (surrogate) posterior samples. In fact, TULiP can be integrated with other logit or predictive distribution-based OOD detectors (EBO, GEN, etc.) by substituting line 18 of Algorithm 1.
>
> **W6: Performance drop on ImageNet-1K.**
>
> As you have pointed out, TULiP does not outperform ASH on ImageNet-1K near/far-OOD setting of Table 1. However, we would like to emphasize that TULiP outperforms ASH by a large margin on both the near- and far-OOD settings with ImageNetV2 weights; see Table 7 of Appendix C.3. Taking both Table 1 and 7 into account, we recognize our method achieves SOTA performance. Note that we have conducted hyper-parameter tuning to ASH for Table 7, as shown in Table 4, but we were unable to achieve good results. One of the reasons is that we have no insights on how the parameter leads ASH to a good performance other than the suggested range in the original paper, since it does not have any theoretical analyses, unlike TULiP. In the main body of the revised manuscript, we will remark on our performance in Table 7. As for the reason of the degrading performance with TULiP in Table 1 (and ASH in Table 7), we are currently investigating as our future work.
>
> **W7: Rearrangement of Table 1.**
>
> Thank you for your thoughtful suggestions. We have rearranged Tables 1 and 2 to better indicate methods requiring training data access.

---

> > ### Comment · Reviewer_enPE · 2024-11-25
> >
> > Thanks for the response.
> >
> > * Regards to GEN [2]: I believe there may be some misunderstanding regarding my comments on GEN. At no point did I claim that TULiP is the same as GEN. Instead, I noted that the OOD score utilized in the paper has already been explored in GEN, which is accurate. Additionally, I highlighted that TULiP differentiates itself by relying on perturbed predictions. Since you mentioned that TULiP can be integrated with other logit- or predictive-distribution-based OOD detectors, it would be beneficial to include results demonstrating this integration.
> > * Regarding performance drop on ImageNet-1K:  If the checkpoint has a significant impact, reporting the average results would be more appropriate. Furthermore, the paper currently lacks a discussion on why performance degrades when using ImageNet-1K as the in-distribution dataset. Including such a discussion would enhance the paper's insights.
> > *  I strongly recommend that the authors carefully develop the related work section to better position their contributions. Additionally, conducting more comprehensive experiments is necessary to support for the claims made in the paper.

---

> ### Author Response · Authors · 2024-11-22
>
> **W8: Alternative architectures such as BiT and ViT.**
>
> We recognize the great success of transformer-based models in recent research. However, they offer a significantly different architecture compared to convolutional networks, with the introduction of self-attention layers, etc. Applying and implementing TULiP to such architectures would require a great effort, and we are still exploring TULiP over a broader spectrum of network architectures in our future research. We will revise our manuscript to emphasize this limitation.
>
> **W9: TULiP and Covariate-shift OOD**
>
> Thank you for pointing that out. In Table 2, the reported metrics are in AUROC.
> It might be true that some compared methods are specified to SS-OOD. However, most of the methods are agnostic to SS-OOD and CS-OOD, especially e.g., MC-Dropout and MDS, which explicitly included experiments in a Covariate-Shift setting in their original work.
>
> TULiP aims to detect covariate-shift OOD samples. To further validate TULiP on CS-OOD, *we have included a new experiment in the revised manuscript Appendix C.4*. We compared TULiP with other UQ methods that are not specified to SS-OOD, using a well-established ImageNet-C Gaussian Blur experiment. The results further suggest TULiP's effectiveness in the CS setting.
>
> Moreover, thank you for mentioning the study [3]. While we saw some relations between our research and [3], let us remark on some differences as follows: the authors of [3] discussed a Semantically Coherent OOD setup, where data with ID labels in the OOD dataset is considered ID. Their proposed method, UDG, involves unsupervised learning with unlabeled OOD samples, i.e., to be precise, different from our problem setting, and is therefore difficult to fairly compare [3] and TULiP. We will mention [3] in the related work section of our revised manuscript.

---

> > ### Comment · Reviewer_enPE · 2024-11-27
> >
> > Regarding W8: If TULiP is only applicable to limited architectures, it would be beneficial to highlight this in the limitations section.
> >
> > Regarding W9: Thank you for clarifying the goal of TULiP. However, I believe there may have been a misunderstanding about my comments on [3]. I explicitly stated:
> >
> > `The purpose of TULiP is somewhat ambiguous—whether it aims to detect covariate-shift OOD samples or to demonstrate robustness against them.  If the LATTER is the case, a fairer comparison would be with methods developed explicitly for covariate-shift OOD samples, such as [3].`

---

> ### Author Response · Authors · 2024-11-27
>
> We would like to thank you again for your prompt and constructive feedback, and for the great effort you put into reviewing our work.
>
> We have uploaded a revision of our manuscript regarding to your thoughtful concerns and invaluable comments.
>
> Regarding to your last reply:
>
> - Regards to GEN [2]: Thank you for your clarification. We have revised Section 2, Table 1 and lines 368-370 **in the revised manuscript** to address the difference more clearly. We have reported results for GEN and TULiP+GEN in the revised Table 1.
> - Performace drop on ImageNet-1K: In **lines 463-473 of the revised paper**, we mentioned the performance drop of TULiP on ImageNet-1K and provided a brief discussion. As you have pointed out, we agree that such analysis will sure be insightful. However, an in-depth (empirical) analysis might be essential to justify any claims and detailed explanations to this phenomena, which leads to an valuable future research direction. Nevertheless, as we stated in the revised paper, AUROC-wise, TULiP still outperforms other baselines (except ASH) by a significant margin.
> - Carefully develop the related work section: We have revised the entire section 2 to better compare our work to existing works and clarify our contributions. We reduced the text amount of UQ methods and assigned more space to position our contributions, also including GEN [2] into the section.
> - Reporting the average results would be more appropriate: Thank you for your kindful suggestion. However, we believe that it is important to follow the standard benchmarks established by Zhang et al. (OpenOOD v1.5), where they have only used the V1 weights, in order to enhance the accessibility and soundness of our work. Therefore, we respectfully keep our current setups with V1 weights in our main table.

---

> ### Author Response · Authors · 2024-11-27
>
> In addition, we briefly summarize the related revisions that we have uploaded, besides the ones mentioned above, as follows:
>
> 1. We have modified the abstract for clarification.
> 2. lines 40-41: As you have pointed out, we emphasized the connection between epistemic uncertainty (EU) and OOD here.
> 3. Slightly simplified Section 3.
> 4. Revised line 272: We have removed the "early checkpoint (optional)" input as 1) Such checkpoints are not tractable under strict post-hoc setting, 2) It leads to confusion and 3) After further investigation, we found that the default setting ($\theta_{t_s} = \mathbf{0}$) achieves sufficient performance gain (see below). We deeply apprepreciate your rightful comments.
>
> Table: CIFAR-10 SS-OOD, with a ResNet-18 trained by SGD momentum=0.9 for 400 epochs (Top-1 acc around 86%).
> | Checkpoints for $\theta_{t_s}$ | near/far AUROC |
> |--|--|
> | No ($\mathbf{0}$) | 81.01 / 81.58 |
> | Yes (Epoch 19) | 81.05 / 81.96 |
>
> 5. Revised lines 244-246: To emphasize our choice taking $t_s = 0$ and $\theta_{t_s} = \mathbf{0}$, we moved the corresponding description from the end of Section 4 to the beginning.
> 6. Revised Table 1 and lines 458-460: We have added GEN as baseline as well as TULiP + GEN. In short, TULiP+GEN boosts GEN performance on CIFAR-10 and ImageNet-1K.
> 7. Revised lines 426-428: We added the citation to [3] and addressed the concerns regarding covariate-shift OOD raised in [3] for enhanced clarity. We believe that this is the appropriate place to address such an issue, rather than Section 2, given that [3] has a different setting compared to our work (as discussed before).
> 8. lines 472-474: We emphasized the results shown in Appendix C.3 (W6).
> 9. Added Appendix C.4: We further investigate the performance of TULiP with ViT-B-16 on ImageNet-1K SS-OOD; please refer to the results in Table 8 of the revised manuscript:
>
> | Method | ImageNet-1K (ViT) SS-OOD (AUROC) |
> | ------ | -------------------------------- |
> |    ViM | 77.03/92.84 |
> |    MDS | 79.04/92.60 |
> |        | ^ Requires training data |
> |    EBO | 62.41/78.98 |
> |    MLS | 68.30/83.54 |
> |    ASH | 53.21/51.56 |
> |    GEN | 76.30/91.35 |
> | TULiP  | 73.63/87.98 |
>
> In summary, representative post-hoc methods (without training data access) fail to perform well overall, with a gap between methods with or without training data access can be observed. It could possibly be because of the significant architectural difference between transformers and CNNs. Please see the discussion in the appendix. We believe that additional exploration is mandatory for future research, perhaps by incorporating architectural knowledge of transformers, as we stated in the revised paper.
>
> We hope that the above explanations will help address your concerns, and we look forward to your response. We sincerely appreciate your constructive comments and your time in reviewing our paper.

---

> > ### Comment · Reviewer_enPE · 2024-11-27
> >
> > Thanks for conducting the suggested experiments.
> > * Based on the results utilizing ViT-B-16 as the backbone with ImageNet-1k as the ID dataset, it seems that TULiP can be regarded as an enhancement method for OOD detection such as ReAct [1] and RankFeat [2].  I suggest that the authors discuss these methods in the related work section to better contextualize TULiP's contribution.
> > * I reviewed Table 8 in Appendix C.4 and noticed that the baseline results were cited from [3]. I’m curious why you didn’t use the reproduced results from your machine, as generating them would only require a single forward pass.
> >
> >
> > [1] ReAct: Out-of-distribution Detection With Rectified Activations. NeurIPS, 2021.
> > [2] RankFeat: Rank-1 Feature Removal for Out-of-distribution Detection. NeurIPS, 2022.
> > [3] Openood v1.5: Enhanced benchmark for out-of-distribution detection. NeurIPS, 2023.

---

> ### Author Response · Authors · 2024-11-28
>
> We appreciate your prompt feedback.
>
> - Regarding W8: Thanks for the suggestion. We have revised line 538 to highlight this limitation.
> - Regarding W9: Thanks for the comment on the study [3]. As you stated, we seem not to understand the comment sufficiently. The reason of our confusion could be that we do not scope to demonstrate the robustness in our manuscript, while the "latter" was emphasized in your response. Would you like to explain more explictly about that comment? We appreciate your clarification.
> As for Table 9 of the current revised manuscript (Appendix C.5, previously C.4) on CS-OOD experiments, we intended to further demonstrate TULiP with baselines not explictly designed for SS-OOD. In your comment, you suggested us a fairer comparision if the "latter" is the case, yet TULiP falls into the "former" category (i.e., aims to detect CS-OOD). Nevertheless, we still believe that this experiment can enhance our paper as we added it, since some baseline methods in Table 2 only considered SS-OOD in their original study as you have rightfully pointed out.
> - ReAct and RankFeat: As you have commented, TULiP is indeed similar to them as an enhancement method that can work with methods in logit and prob spaces. Therefore, we have added them to line 98.
> - Why we didn’t use the reproduced results from our machine: During our discussion, we first conducted the experiment using GEN+ViT-B-16, and we obtained the exact same result as being listed in OpenOOD repository [1] since we used the same pretrained weights provided by torchvision. Therefore, we simply cited other baseline results as it will still take shorter time, even though only a single forward pass is required.
>
> [1] OpenOOD v1.5 results. https://docs.google.com/spreadsheets/d/1mTFrO-_STYBRcNMMEmHQrFPQzeg6S8Z2vRA8jawTwBw/edit?usp=sharing (Last accessed 2024-11-28 04:04 GMT+0).
> [3] Semantically Coherent Out-of-Distribution Detection. In ICCV, 2023.
>
> Once again, we are deeply grateful for your invaluable time and effort in reviewing our paper, and we would greatly appreciate your feedback.

---

### Official Review · Reviewer_3GdG · 2024-11-02

**Soundness:** 3
**Presentation:** 3
**Contribution:** 3
**Rating:** 6
**Confidence:** 3

**Summary:**

This paper proposes a novel uncertainty estimation method TULiP for OOD detection. The core idea of the paper is to generate uncertainty scores by perturbing model parameters based on linearized training dynamics.

**Strengths:**

* The paper is clearly written and experiments have been extensively conducted across a set of diverse datasets.

* Theoretical analysis is thorough.

**Weaknesses:**

* There are so many hyperparameters that implementing the method in realistic scenario may have some difficulties.

* Performance on far OOD is not good enough.

**Questions:**

How does the performance compared to deep ensembles[1]？

[1] Simple and scalable predictive uncertainty estimation using deep ensembles

---

> ### Author Response · Authors · 2024-11-22
>
> First of all, we thank the reviewer for reviewing the manuscript and acknowledging our contributions. We provide additional clarification and explanations for your concerns.
>
> We uploaded a revision of our manuscript. In the following, unless otherwise mentioned, we refer to sections and figures/tables in our original manuscript.
>
> **W1: There are so many hyperparameters that implementing the method in a realistic scenario may have some difficulties.**
>
> Indeed, TULiP involves $\epsilon, \lambda, \delta$ as its hyper-parameters. It is natural that one considers it difficult to adopt for real use cases. However, we have found that the parameter optimization process for TULiP is notably manageable.
>
> In Appendix C.3 and Table 7, we demonstrate the efficiency of the hyper-parameter tuning of TULiP, using torchvision's ImageNetV2 weights. The same hyper-parameter search range, as shown in Table 4, was used. Despite the increased number of hyperparams, TULiP achieved top results (note that ViM requires access to training data) in this challenging scenario, where all methods suffer from a significant performance drop due to the nature of V2 weights (further discussed in Appendix C.3).
>
> When the validation sets are relatively small, it usually takes ~10mins to search an optimal hyper-parameter set of TULiP. If the time is constrained, $\epsilon$ is the most important parameter (line 501, 1331), and solely tuning on it while fixing a reasonable $\lambda, \delta$ (e.g., suggested value in line 418) usually yields good results (cf. Figure 4). We will add the discussion to our revised manuscript.
>
> **W2: Performance on far OOD is not good enough.**
>
> As you have pointed out, TULiP's performance on far-OOD is not as good as its near-OOD results within Table 1. In the table, TULiP is mainly outperformed by either ViM or ASH. However, ViM requires access to training data, which is different from our setup and accessed more information, while ASH is significantly unstable in some cases; see the last row of Table 7 in Appendix C.3. Based on both Table 1 and 7, we consider that TULiP is on par with ASH on the far-OOD scenario. We will add a remark on our far-OOD performance in Table 7 in the main body of our revised manuscript.
>
> Note that ASH has no theoretical insights and, therefore, lacks explainability and relies more on heuristics to tune with. This could potentially be one of the reasons that ASH fails in the experiment in Appendix C.3.

---

> ### Author Response · Authors · 2024-11-22
>
> **Q1: Comparison with Deep Ensembles (DE)**
>
> Due to the high computational cost of DE, we are unable to evaluate it under our experiment setup. However, we do have the following comparison on the near-OOD setup of CIFAR-10 and CIFAR-100, following [1]:
>
> | | CIFAR10-near | CIFAR100-near |
> |-|---|---|
> |DE|90.6|82.7|
> |TULiP|89.7|81.3|
>
> Under this setup, it is clear that TULiP does not fall much behind Deep Ensembles, despite the restricted access to various information (e.g., training process, training data, etc.) and a much smaller computational overhead.
>
> Furthermore, *in the revised manuscript Appendix C.4*, we report experiment results under the setting of [2], where TULiP outperforms Deep Ensemble in this covariate-shift scenario.
>
> [1]. OpenOOD: Benchmarking Generalized Out-of-Distribution Detection, NeurIPS'22 Datasets and Benchmarks
> [2]. Single model uncertainty estimation via stochastic data centering, NeurIPS'22

---

> > ### Comment · Reviewer_3GdG · 2024-11-26
> >
> > Thanks to the authors' response.
> >
> > My concerns have been addressed and I will keep my rating

---

> ### Author Response · Authors · 2024-11-27
>
> We would like to thank you again for your great effort on reviewing our work and your appreciation. We are happy that your concerns have been addressed.
>
> We have uploaded a revision of our manuscript regarding to your thoughtful concerns and invaluable comments.
> We briefly summarize the related revisions as follows:
>
> 1. Revised Appendix B.2: Added discussion on determining hyper-parameters in practice (W1).
> 2. line 462: We emphasized the results shown in Appendix C.3 (W2).
>
> We sincerely appreciate your time in reviewing our paper. If you have any questions or concerns about our work, we are glad to provide more discussions.

---

### Official Review · Reviewer_mnne · 2024-11-03

**Soundness:** 2
**Presentation:** 2
**Contribution:** 2
**Rating:** 6
**Confidence:** 2

**Summary:**

This paper proposes a test-time post-hoc OOD detection method, which is theoretically driven by considering hypothetical perturbations applied to model parameters before convergence, allowing for the computation of an uncertainty score. The overall idea is interesting. However, there are a few points to be clarified.

**Strengths:**

This paper is overall well-written. The idea is theoretically driven and offers good interpretability. The method is thoroughly evaluated and demonstrates excellent performance compared to many OOD detection methods.

**Weaknesses:**

1.What's the motivation for calculating the upperbound of variations for uncertainty quantification? As shown in Eq 1. The objective is to estimate the variance given an different parameters initializations. To solve this, the DNN is first linearized locally with the NTK theory and the upperbound for introducing the changes are calculated with the NTK theory.  The paradox is if the parameters can be already be perturbed, why NTK is needed for calculating the upperbound. Besides, calculating the upperbound will bring biased estimations of uncertainty. Another simple way to achieve this might be directly apply random perturbations to the network parameters (like random noises injection, dropout parameters), can easily get ensemble of neural network parameters. What is the advantage over these methods?

2. Given that $\lambda \in\{\sqrt{o}, 3 \sqrt{o}\}$, where $o$ represents the number of output dimensions, why does Figure 4 only explore the range of $\lambda$ values between 0 and 3 on ImageNet-200? The authors should consider exploring a broader range of this hyperparameter.

3. The authors mention that TULiP is over three times faster than ViM, noting that ViM takes more than 30 minutes just to extract ID information on a recent GPU machine. However, it appears that the proposed method requires $M=10$ forward passes per sample for OOD detection. Compared to classic OOD detectors like EBO, does this imply that the detection speed of the proposed method is relatively slower?

4. In the experiments, the authors calculated Equation 8 using 256 samples from the ID dataset (ImageNet-1K) and 128 samples per OOD dataset. However, the authors do not clarify how these 256 ID samples and 128 OOD samples were selected or whether OOD samples align with test samples. Additionally, did the authors know beforehand which samples were ID and OOD when using these samples?

6. Have the authors considered the impact of different types of OOD data? For example, have the authors considered situations where OOD data is very far from ID data to improve detection of far-OOD.

7.Why can Equation 11 be approximated by this way proposed by the authors? This approximation ($\nabla_{\boldsymbol{\theta}} f_T^{emp}(\boldsymbol{z}) \boldsymbol{\Gamma} \approx \nabla_{\boldsymbol{\theta}} f(\boldsymbol{z})$) only considers the impact of params in each layer and does not account for the effect of the order of layers with the same params in network. Figure 1: a) Although it presents training trajectories under different params, it does not indicate which specific layer each color represents. The authors should conduct more experiments and theoretical analyses to explore this aspect.

**Questions:**

See weakness.

---

> ### Author Response · Authors · 2024-11-22
>
> We appreciate your review of our manuscript and valuable feedback, especially on the theoretical aspects. We will provide additional clarifications and address your concerns here.
>
> We uploaded a revision of our manuscript. In the following, unless otherwise mentioned, we refer to sections and figures/tables in our original manuscript.
>
> **W1: What's the motivation for calculating the upperbound of variations for uncertainty quantification?**
>
> Firstly, in our post-hoc setting, we have no access to training data, process, and model parameters prior to convergence; we only have a trained, converged model. In such a case, direct computation of Eq. (1) is impossible. Furthermore, the perturbation in Theorem 3.1 is hypothetical (line 154). As we explained around lines 150-154, perturbation to model parameters prior to convergence is also intractable in a post-hoc setting since it requires re-training. Therefore, a bound like Eq. (5) is derived to overcome such difficulty. More importantly, our method addresses the uncertainty caused by the training process (lines 133-135) as it is critical (lines 43-50) to uncertainty estimation.
>
> Deep Ensembles [1] computes Eq. (1) in its exact form. In *Appendix C.4 of the revised manuscript*, we provide additional experiments comparing our method and Deep Ensembles, alongside other methods.
>
> We also note that our perturbation scheme, hence Theorem 3.1, links back to Eq. (1) in the following sense:
>
> In our theoretical framework, we consider the infinite width limit under the NTK scaling [2]. Under such limit, the empirical NTK (at initialization) converges to a specific deterministic kernel $\Theta$, where the distribution of a neural network $f(x; \theta)$'s initialization functional $f_\mathrm{Init}(x)$ converges to a Gaussian Process (NNGP). In Eq. (2), it is equalivent to a deterministic (fixed) $\left. \nabla_\theta f_\mathrm{True}(x) \right|\_{\theta = \theta^\ast}$ and a stochastic $f_\mathrm{Init}$ following the NNGP.
>
> Using the model defined in Eq. (2) and the training process described in Eq. (3), Eq. (1) effectively becomes:
>
> $\mathrm{Var}\_{f_\mathrm{Init} \sim \mu_\mathrm{NNGP}}[f_T(x;\theta | \mathrm{Init} = f_\mathrm{Init})],$
>
> where $f_T(x;\theta | \mathrm{Init} = f_\mathrm{Init})$ indicates a network trained via Eq. (3) by time $T$, with $f_\mathrm{Init}$ as initialization.
>
> When we set $t_s = 0$ (the initialization time), the perturbation $\Delta f$ will be applied to $f_\mathrm{Init}$. Therefore, given a fixed initialization $f_0$ to perturb, Eq. (5) computes an upper-bound over a perturbation of the initialization functional:
>
> $\mathrm{Var}_{\Delta f}[f_T(x;\theta|\mathrm{Init} = f_0 + \Delta f)],$
>
> since $\hat{f}_T$ is supposed to be trained from initialization $f_0 + \Delta f$, we have $\hat{f}_T = f_T(x;\theta | \mathrm{Init} = f_0 + \Delta f)$, hence the above.
>
> Comparing it to the previous equation, we see that the difference between them is the distribution of the initialization functional $f_\mathrm{Init}$. In Eq. (1), $f_\mathrm{Init}$ comes from the NNGP; while in Eq. (5), it is centered around $f_0$ with a stochastic perturbation $\Delta f$. Intuitively, by using Eq. (5), we are approximating the predictive variance trained from the NNGP prior with the predictive variance trained from a random perturbed initialization $f_0 + \Delta f$. Figure 2 demonstrates the effectiveness of such an approximation. Notably, TULiP is tractable under our post-hoc setting, whereas Eq. (1) is not and typically requires tremendous computational effort.
>
> We will add above discussion to our revised manuscript.
>
> [1] Simple and Scalable Predictive Uncertainty Estimation using Deep Ensembles, NeurIPS'17
> [2] Neural tangent kernel: Convergence and generalization in neural networks, NeurIPS'18

---

> ### Author Response · Authors · 2024-11-22
>
> **W2: Why does Figure 4 only explore the range of $\lambda$ values between 0 and 3 on ImageNet-200?**
>
> We would like to clarify that in Figure 4, the value of $\lambda$ should be read as e.g., $2 \sqrt{o}$ when the horizontal tick or label shows $2$. Therefore, in Figure 4, the considered range of $\lambda$ is $0 \sim 3 \sqrt{o}$. In Figure 4 c) d), $\lambda = 1.5 \sqrt{o}$ is considered as it yields better performance on ImageNet-200. However, to keep a relatively small hparam range for easy tuning on a validation set, we have suggested the range $\lambda \in \sqrt{o}, 3\sqrt{o}$ and reported the results in our tables based on such choice.
>
> **W3: Concerns regarding computational efficiency.**
>
> As you have pointed out, TULiP requires $\mathcal{O}(M)$ forward passes. It is true that TULiP is slow in this sense. Nevertheless, forward passes are relatively cheaper, which does not necessarily mean TULiP is 10 times slower than, e.g., EBO. TULiP also gives better performance, as shown in Table 1, at the cost of computational efficiency, yet tractable under post-hoc scenarios. We will add the discussion to our revised manuscript.
>
> **W4: ID / OOD setups, especially in Figure 1.d).**
>
> In all of our experiments, we consider the ID and corresponding OOD sets from the dataset pairings described in Section 5.2 (lines 446-453, 459-464). Since OOD datasets are manually selected for a given ID dataset, we know beforehand whether it is ID or OOD for a datapoint. We will revise the manuscript to improve the clarity. For Figure 1 d), all samples are uniformly sampled from corresponding datasets, and color represents its dataset (406-411).
>
> **W5: Impact of different types of OOD data.**
>
> In this work, we did not consider special cases, e.g., very-far OOD data. Considering those special cases might indeed be beneficial for far-OOD performance, and it could be a promising research direction for future work.
>
> In our current work, we aim to develop a theoretically-driven post-hoc OOD detector that works for all kinds of OOD data, either near-ID or far-ID. Indeed, in Table 1, TULiP's performance on far-OOD is by no means bad since ViM requires training data access and ASH comes with no theory and is sometimes unstable (e.g., Appendix C.3). At the same time, TULiP achieves remarkable result on near-OOD settings.

---

> ### Author Response · Authors · 2024-11-22
>
> **W7: Approximation of the true NTK**
>
> Thank you for pointing out this important aspect.
> First, in Figure 1 a-c), the color represents the number of parameters in each convolutional layer, with blue corresponding to layers with more parameters and green to those with fewer. Only one network was trained to produce all three parts of Figure 1 a-c).
>
> Regarding the following question:
>
> > This approximation only considers the impact of parameters in each layer and does not account for the effect of the order of layers with the same parameters in the network.
>
> If this question implies that the approximation in Eq. (11) does not account for the interplay between layers in deep neural networks, we acknowledge your concern. Indeed, our scaling factor $\Gamma$ is based solely on the number of parameters in the layers, potentially ignoring their order and inter-layer dynamics.
>
> When the neural network has only a single hidden layer, the NTK can be well-approximated using Eq. (11) based on the definition of the NTK. However, as demonstrated in our numerical experiments (Section 5), our approach performs well even without explicitly considering inter-layer relationships, relying only on the number of parameters in the layers. We chose this approach for its simplicity, as investigating inter-layer dynamics would require significant additional effort.
>
> As you have kindly pointed out, further investigation is indeed necessary for deep neural networks. We will revise the manuscript to highlight this limitation more explicitly.
>
> If we have misunderstood your question, please let us know, and we will reply accordingly.

---

> ### Comment · Reviewer_mnne · 2024-11-27
> **Further comments**
>
> Thank you very much for the detailed responses. The overall idea is interesting. After carefully reading the revised paper again, there are a few questions:
> 1. In line 33, it says "Under our problem setting, neither the distribution of initialized models nor the training process is accessible, .......... perturbation applied towards the network function f(x), at a time t = ts before the training terminates at t = T."
>
> This seems to be contradict with each other, as it requires the model before convergence, which means the training process is accessible.
>
> Besides, if you can access the training process, why cannot access the distribution of initialized models. Besides, the initialization shall be known random distributions.
>
> 2. If the (partial) training process can be accessed (at time t= ts), it would not be hard to know \hat{f}_{T}(z), then the bound of differences ||f_{T}(z) - \hat{f}_{T}(z)|| should be known. Then, we might not need such sophisticated methods for bounding this term.
>
> 3.A bit more explanations on why Eq 6 holds. Though some people can see this can be achieved by expanding the left term, a bit more explanations can help more people to understand this equation.
>
> 4.Following above the questions, can the author list some practical scenarios, the settings in the paper will happen?
>
> Overall, the math is clean and neat in this paper which shall be appreciated. However, the reviewer is unsure about the motivation  for this work. Maybe this work can be reformulated as some methods for uncertainty estimation, like what the dropout as Bayesian method paper will do.  Please correct me if I misunderstood anything. Thanks again for the responses.

---

> ### Author Response · Authors · 2024-11-27
>
> We appreciate your response and your regonization towards our theortical contributions. Thank you again for your great effort on reviewing our work.
>
> We have uploaded a revision of our manuscript regarding to your thoughtful concerns and invaluable comments.
>
> Regards to your last reply:
>
> - It is correct that, as you have said, in our setting neither the distribution of initialized models nor the training process is accessible. **We have no access to the training process and therefore we did not perturb the network prior to convergence in practice** (lines 152-156). In fact, the functional perturbation prior to convergence is only for establishing the theortical framework, ultimately leads us to our bound (Thm 3.1, Prop 3.3), and a practical method (TULiP) to evaluate this bound. In practice, since we don't know $t_s$ or $\theta_{t_s}$, to estimate the bound we have assumed $t_s = 0$ and $\theta_{t_s} \approx \mathbb{E} \theta_{t_s} = \mathbf{0}$. In our revised manuscript, we have emphasized this in lines 137-139.
> - For Eq. 6, thank you for your comments. We have revised our paper with a remark in line 194 and lines 989-992.
> - For practical scenarios: As we address your concerns above, it might be clear that our work is potentially broadly applicable in practice, for example, autonomous driving and medical applications (line 29). TULiP works in a post-hoc setting, where only the trained parameters are required to apply our method, without access to training data and training process. Post-hoc methods are widely used in practice (lines 42-47).
>
> We hope that the above explanations will provide you a clearer intuition of our work and motivation.
>
>
> Below, we briefly summarize the revisions related to your previous concerns as follows:
>
> 1. Added Appendix A.7: Discussion about connection between Eq. 5 and Eq. 1 (W1).
> 2. Added Appendix C.6: Discussion regarding computational efficiency (W3), where we have included wall-clock time comparision between TULiP and single forward-pass methods (EBO).
> 3. Revised line 417: Clarification of ID/OOD setups (W4).
> 4. Revised line 264, 535: Highlighting the limitation of current layer-wise scaling scheme (W7).
> 5. Appendix C.2 (Added **Figure 5**): We added an additional empirical evidence justifying our layer-wise scaling scheme, where the scaled NTK and in-training **empirical NTK is directly computed and compared** using training data (W7). Please refer to Figure 5 (line 1361) for details.
>
> We hope that above explanations will help address your concerns, as we look forward to your response. We sincerely appreciate your constructive comments and your time in reviewing our paper.

---

> > ### Comment · Reviewer_mnne · 2024-12-02
> >
> > Thank you for your  further clarifications. I will raise my score. But the example you mentioned can be more concrete, like how in detail the method can be used for autonomous driving and medical applications.

---

> ### Author Response · Authors · 2024-12-03
>
> We greatly appreciate your graceful recognition of our paper and your kind response.
> We will provide additional discussion below, regarding your recent question.
>
> For example, computer vision tasks like object detection (OD) and semantic segmentation (SS) are fundamental for autonomous driving, as they are the core building blocks to "the eyes" of an autonomous vehicle. As we noted in our paper, its crucial to know when the input is far from the training set (being OOD) (for OD, see [1]; for SS, see [2]), as deep NNs sometimes provides over-confident predictions, raising safety concerns. Post-hoc OOD detectors (including TULiP) can be easily attached to trained models (OD: mixed regression + classification, SS: classification), help the model to identify and prevent it from unwanted, potentially unsafe predictions.
>
> As shown in above, being post-hoc (like TULiP) broadens the application of a method as it can be applied to pre-trained models. We will futher revise our manuscript by describing the practical scenario in detail.
>
> [1]. Unknown-Aware Object Detection: Learning What You Don’t Know from Videos in the Wild, CVPR'22
> [2]. Entropy Maximization and Meta Classification for Out-of-Distribution Detection in Semantic Segmentation, CVPR'21

---

### Meta-Review · Area_Chair_gQW9 · 2024-12-17

**Metareview:**

This paper introduces TULiP, a test-time uncertainty estimation method designed for out-of-distribution (OOD) detection. TULiP leverages a linearized training dynamics framework based on the Neural Tangent Kernel (NTK) to approximate uncertainty at test time. Specifically, the method perturbs the model weights in the linearized space to compute predictive uncertainties, which are then used to distinguish OOD samples from in-distribution (ID) data.

While TULiP operates at test time, its reliance on training assumptions disqualifies it as a true post-hoc OOD detector.
The authors should reframe their claims and emphasize that TULiP is a training-dependent uncertainty estimation method rather than a universally applicable post-hoc OOD detector.

Therefore, it is recommended that the paper be revised and resubmitted in a future version. The authors should carefully address the reviewers' comments provided throughout the review process. Below are additional comments from Reviewer enPE after the rebuttal.

**Additional Comments On Reviewer Discussion:**

Reviewer enPE has primary concerns regarding the positioning of this paper's contributions and the absence of two critical comparative studies.

- While TULiP establishes a theoretical framework, its implementation involves adding Gaussian noise to the pre-trained checkpoint to create multiple "pseudo" ensemble models. Additionally, the “early checkpoint” in Algorithm 1 defaults to zero (Line 376), despite being labeled optional. This creates a disconnect between the theoretical foundation and the practical implementation, which makes it difficult for me to be convinced that the proposed theory sufficiently supports the implementation.

- By injecting Gaussian noise into the weights of the pre-trained checkpoint, TULiP generates varying predictive distributions, which are then combined with different OOD score functions. Although TULiP is presented as a post-hoc OOD detector, I believe it more appropriately belongs in the category of enhancement methods. The authors appear to agree with this perspective. A fair evaluation would require comparisons with other enhancement methods, such as ReAct [1] and RankFeat [2].

- The utility of TULiP is demonstrated through its application to OOD detection. However, its deployment may face significant limitations since it requires access to the weight parameters of each layer. Furthermore, TULiP's performance is inferior in far-OOD scenarios.

---

### Decision · Program_Chairs · 2025-01-22

Reject